# Resounding Acoustic Fields with Reciprocity

**Zitong Lan**
University of Pennsylvania
ztlan@seas.upenn.edu

**Yiduo Hao**
University of Pennsylvania
yiduohao@seas.upenn.edu

**Mingmin Zhao**
University of Pennsylvania
mingminz@seas.upenn.edu

https://waves.seas.upenn.edu/projects/Versa

## Abstract

Achieving immersive auditory experiences in virtual environments requires flexible sound modeling that supports dynamic source positions. In this paper, we introduce a task called resounding, which aims to estimate room impulse responses at arbitrary emitter location from a sparse set of measured emitter positions, analogous to the relighting problem in vision. We leverage the reciprocity property and introduce Versa, a physics-inspired approach to facilitating acoustic field learning. Our method creates physically valid samples with dense virtual emitter positions by exchanging emitter and listener poses. We also identify challenges in deploying reciprocity due to emitter/listener gain patterns and propose a self-supervised learning approach to address them. Results show that Versa substantially improve the performance of acoustic field learning on both simulated and real-world datasets across different metrics. Perceptual user studies show that Versa can greatly improve the immersive spatial sound experience. Code, dataset and demo videos are available on the project website.

## 1  Introduction

The rapid development of AR/VR technology has highlighted the role of multi-sensory synthesis, where both visual and audio signal must be rendered coherently to provide immersive experiences [11, 42, 53, 64]. While significant progress has been made in modeling dynamic visual field [17, 20, 63], the acoustic counterpart remains largely limited to static scenarios. Recent works have demonstrated impressive results in modeling acoustic fields [8, 34, 37, 39, 44–46, 54, 59], but primarily focus on a fixed sound source. This constraint limits auditory experiences where sources move through the scene, such as conversations between walking participants or footsteps echoing through corridors.

In this paper, we study a novel task of estimating acoustic fields at arbitrary emitter positions, given observations from only a *sparse* (i.e., fewer than 10) set of emitters. This setup is motivated by practical deployment constraints: while microphones are compact, inexpensive, and easy to place in dense arrays, speakers are bulky, power-hungry, and challenging to install at scale [18]. Moreover, simultaneously operating multiple speakers in the same room can introduce significant interference. We frame this task as *resounding*, parallel to *relighting* in computer graphics. Just as relighting enables control over illumination in virtual scenes, resounding enables dynamic placement of sound sources while preserving physical realism. The core challenge lies in modeling how acoustic fields vary with emitter positions in a way that is physically consistent and generalizable.

Modeling acoustic fields fundamentally relies on estimating impulse responses, which characterize how sound waves propagate within an environment. Existing learning-based approaches to this problem can be broadly categorized into two classes. The first class follows the neural radiance field paradigm [41] in graphics, where neural networks learn spatially continuous acoustic fields from data [34, 36, 37, 39, 54]. However, these methods struggle with sparse emitter locations as they require densely (i.e., hundreds or thousands) deployed emitters to generalize to novel emitter locations. The second class leverages differentiable ray tracing [59, 26] to explicitly model acoustic propagation paths. While this approach improves generalization, it relies heavily on simplified geometry (e.g., a few dozen planar surfaces), leading to significant errors in environments with complex structures.

39th Conference on Neural Information Processing Systems (NeurIPS 2025).

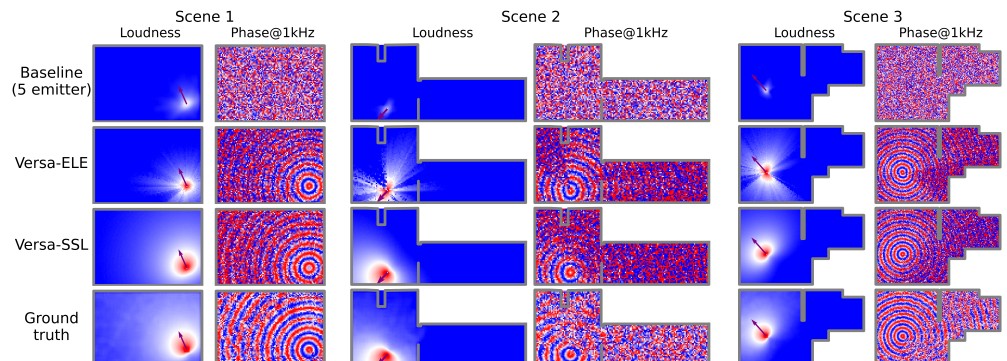

**Figure 1: Estimated acoustic field at novel emitter position.** For simulated scenes (*five* training emitters each), we show loudness distribution (red: loud, blue: quiet) and phase patterns at 1 kHz (periodic blue-red). Purple arrows mark emitter poses. Compared to the baseline (AVR [34]), Versa-ELE improves phase map, and Versa-SSL achieves accurate energy map with proper directivity.

In this paper, we introduce Versa, a physics-inspired approach that facilitates realistic resounding under sparse emitter configurations. Our method draws inspiration from *reciprocity*, a fundamental principle in wave propagation. Reciprocity states that if the roles of an emitter and a receiver are exchanged, the wave traverses the same path *in reverse*, and the cumulative propagation effects remain *unchanged*. This principle holds across various wave phenomena, including light [52], radio-frequency signals [6], and acoustic waves [31]. In graphics, this principle underpins powerful algorithms like bidirectional path tracing [32]. Similarly, in acoustics, for any path between source and receiver, exchanging their positions preserves wave propagation characteristics [31, 48, 57].

Building on the principle of reciprocity, we propose Emitter Listener Exchange (Versa-ELE), a strategy that generates physically valid virtual training samples by swapping the roles of emitters and listeners while enforcing the same impulse response. This technique addresses a key asymmetry in data collection: dense sampling of listener positions is relatively easier due to the compact and low-cost nature of microphones, whereas emitter deployment is far more limited due to cost, interference, and physical constraints. Versa-ELE effectively transforms densely placed microphones into dense virtual speakers. We implement Versa-ELE as data augmentation and demonstrate that it consistently improves performance across a range of acoustic field models.

While Versa-ELE is broadly effective, it assumes same directional patterns between emitters and listeners to ensure that the exchanged impulse responses remain unchanged. In practice, however, real-world devices can exhibit asymmetric directional gain patterns, such as an omnidirectional speaker versus a highly directional shotgun microphone. To address this limitation without discarding the core insight of reciprocity, we introduce a complementary Self-Supervised Learning strategy (Versa-SSL) that incorporates reciprocity as a constraint on the model's predictions. Our key idea is to decouple directional patterns from the acoustic propagation effects, allowing the model to learn from swapped emitter/listener pairs by enforcing consistency between their predicted outputs. This approach generalizes reciprocity to a broader set of scenarios and leads to more robust and physically grounded acoustic field estimation, as shown in Fig. 1.

We comprehensively evaluate Versa on the resounding task using both simulated and real-world datasets. Versa-ELE is model-agnostic and can be applied to existing neural acoustic field models, yielding average improvements of 34% on C50 and 31% on STFT. On top of the AVR [34], Versa-SSL further improves performance by 24% on C50 and 48% on STFT, demonstrating its effectiveness in scenarios with asymmetric gain patterns. Perceptual study additionally confirms that Versa enhances spatial audio realism and directional consistency.

In summary, our work makes the following contributions:

- We leverage reciprocity in wave propagation and propose Versa-ELE, a simple yet effective strategy to augment sparse emitter data by generating physically valid virtual samples.
- We introduce Versa-SSL, a self-supervised learning framework that enforces reciprocity-based consistency in model predictions and generalizes to asymmetric directional gain patterns. These methods serve as a general machine learning training strategy grounded in physical reciprocity.
- We implement Versa on multiple models with comprehensive evaluations. Results demonstrate Versa significantly improves acoustic field estimation and enables perceptually realistic resounding.

## 2  Related Work

**Room Impulse Response Modeling.** Impulse response modeling has recently gained a lot of attention, driven by advances in neural acoustic field learning and immersive audio rendering for virtual environments. Early methods [5, 40, 58] relied on audio codec and spatial interpolation, which incurred high memory costs and limited inference fidelity [18, 39]. Recent works [8, 13, 26, 34, 36, 37, 39, 54] employ neural acoustic fields that directly map emitter/listener poses to impulse response and achieve better performance. However, these methods require dense emitters to achieve resounding. Our proposed Versa serves as an add-on to achieve better resounding performance on these methods. In addition, several works have investigated acoustic adaptation to novel scenes [12, 14, 38]. Although generalization to unseen environments is beyond our current scope, our reciprocity-inspired approach can be naturally integrated into both their training and inference pipelines.

**Virtual Experience.** Novel view synthesis and realistic relighting techniques have enabled many immersive virtual experiences. Neural Radiance Fields (NeRF) [41] and Gaussian Splatting [27] have transformed the ability to synthesize highly realistic novel view images from sparse inputs. More recent works, including relighting-focused variants [63, 17, 20, 22], extend these methods to dynamic lighting conditions. Our resounding task, analogous to the relighting problem, aims to enhance flexibility in audio experiences [33, 34, 62].

**PINN for Acoustic Modeling.** Physics-Informed Neural Networks (PINNs) [9, 30, 49, 50] solve the Helmholtz equation directly within a neural framework to estimate acoustic fields from measured impulse responses. These methods learn an implicit representation of the underlying PDE parameters without requiring an explicit mesh, but they usually require dense sampling to enforce boundary conditions. In contrast, our method incorporates the physics of acoustic propagation in the training, improving both efficiency and robustness of neural acoustic field estimation.

**Self-Supervised Learning.** SSL has by now become a staple of feature representation learning for computer vision [60, 15, 25, 16], audio learning [19, 3, 47, 2], and other multi-modal learning tasks [43, 1, 24, 61]. SSL relies on augmentation to create synthetic positive pairs to enforce semantic invariance in the trained model, represented by contrastive methods like SimCLR [25] and so on [1, 3]. Different from prior SSL methods that learn feature representations, our work enforces consistent impulse response outputs to make the model follow signal propagation physics.

## 3  Method

Impulse response is the summation of the sound propagating through multiple paths, including direct path, early reflections and late reverberations. We first describe the impulse response reciprocity with a single path, then extend it to multiple paths. Finally, we introduce our method to improve impulse response estimation inspired by reciprocity via exchanging emitter and listener poses.

### 3.1  Impulse Response and Acoustic Reciprocity

**Single-Path Impulse Response.** We model the acoustic impulse response with a ray-based Geometric Acoustic (GA) model similar to [34, 59]. We consider first sound propagation between an emitter located at $p_e$ and a listener located at $p_l$ along a single path $\mathcal{P}$. This path $\mathcal{P}$ is defined as a sequence of points: $\mathcal{P} = \{p_0 = p_e, p_1, p_2, \ldots, p_K, p_{K+1} = p_l\}$. For each consecutive pair of points $p_k$ and $p_{k+1}$ along the path, we define $\omega_k$ as the direction from $p_k$ to $p_{k+1}$. We define the impulse response $h(t; \mathcal{P}, \omega_e, \omega_l)$ for this single path, which characterizes the sound received at $p_l$ when the emitter at $p_e$ sends out a pulse (i.e., a Dirac delta function), with $\omega_e$ and $\omega_l$ being the orientations of emitter and listener. The response is influenced by the gain patterns of the emitter and listener as well as the effects of sound propagation along the path:

$$h(t; \mathcal{P}, \omega_e, \omega_l) = G_e(\omega_0; \omega_e) \, \Gamma(t; \mathcal{P}) \, G_l(\omega_K; \omega_l), \tag{1}$$

where $G_e(\cdot)$ and $G_l(\cdot)$ represent the gain patterns of the emitter and listener, respectively. $\Gamma(t; \mathcal{P})$ denotes the *path impact function* for the path $\mathcal{P}$, as described below. Note that for any dry source signal $s(t)$ emitted at $p_e$, the listener at $p_l$ receives the reverberant signal $y(t) = s(t) * h(t; \mathcal{P}, \omega_e, \omega_l)$, i.e., the convolution of the dry signal with the path-dependent impulse response $h$.

**Path Impact Function.** $\Gamma(t; \mathcal{P})$ is a time signal that represents the impulse response along a single propagation path $\mathcal{P}$. It characterizes the propagation effect [51] (both attenuation and time delay) in

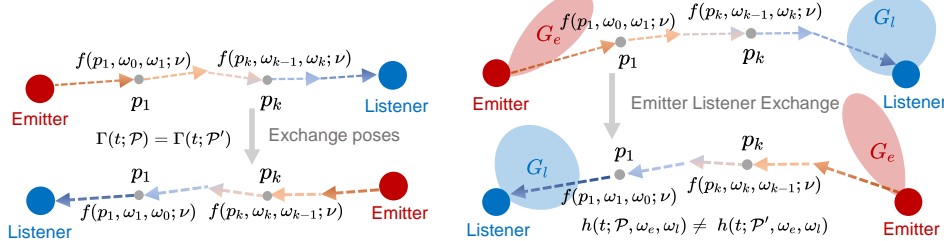

**Figure 2: Left: Reciprocity in acoustic propagation.** The path impact function $\Gamma(\cdot)$ is invariant to swapping the emitter and listener poses, because the local acoustic transfer function $f(\cdot)$ at each point is unchanged when the incident and outgoing signal directions are reversed. **Right: Impact of gain patterns.** Although the propagation path and path impact function remain the same, differences in emitter and listener gain patterns $G$ produce distinct impulse responses $h(t; \cdot)$.

response to a Dirac delta function:

$$\Gamma(t; \mathcal{P}) = \frac{1}{d_{\mathcal{P}}} \delta\left(t - \frac{d_{\mathcal{P}}}{c}\right) * \mathcal{F}^{-1}\left(\prod_{k=1}^{K} f\left(p_k, \omega_{k-1}, \omega_k; \nu\right)\right). \tag{2}$$

The right-hand side of the equation consists of three components. **1)** $\frac{1}{d_{\mathcal{P}}}$ models the attenuation due to wave propagation, where $d_{\mathcal{P}}$ is the total travel distance along path $\mathcal{P}$. This term captures the free-space propagation loss, accounting for the energy spreading as the wave travels through space. **2)** The time delay is modeled by the shifted delta function $\delta(t - \frac{d_{\mathcal{P}}}{c})$, where $c$ is the speed of sound. It indicates that the signal arrives only after traveling for time $\frac{d_{\mathcal{P}}}{c}$ along the path. **3)** $f(p_k, \omega_{k-1}, \omega_k; \nu)$ is an acoustic transfer function modeling how the signal changes when interacting with a surface at point $p_k$, given the incoming direction $\omega_{k-1}$, outgoing direction $\omega_k$, and frequency $\nu$. These interactions include reflection, diffraction, scattering, and transmission. The overall frequency-dependent attenuation due to environmental interactions is captured by $\prod_{k=1}^{K} f(p_k, \omega_{k-1}, \omega_k; \nu)$, where the product accumulates the effect of all $K$ interactions along the path. The inverse Fourier Transform $\mathcal{F}^{-1}(\cdot)$ converts this frequency-domain response into the time domain. Finally, the convolution operation $*$ models how the propagation-related terms are modulated by the frequency-dependent interaction effects.

**Reciprocity in a Single Path.** Reciprocity in acoustic propagation at an interaction surface (similar to BRDF in light transport [21]) implies that the transferred energy remains consistent when the signal direction is reversed: $f(p_k, \omega_{k-1}, \omega_k; \nu) = f(p_k, \omega_k, \omega_{k-1}; \nu)$. Since each interaction along the path is reversible, the overall signal propagation along the whole path is also reversible. If the emitter and listener poses are exchanged, the path impact function remains the same (left of Fig. 2) along the reversed path $\mathcal{P}' = \{p_{K+1} = p_e, p_K, \ldots, p_1, p_0 = p_l\}$:

$$\begin{aligned}
\Gamma(t; \mathcal{P}') &= \frac{1}{d_{\mathcal{P}}} \delta\left(t - \frac{d_{\mathcal{P}}}{c}\right) * \mathcal{F}^{-1}\left(\prod_{k=1}^{K} f\left(p_k, \omega_k, \omega_{k-1}; \nu\right)\right) \\
&= \frac{1}{d_{\mathcal{P}}} \delta\left(t - \frac{d_{\mathcal{P}}}{c}\right) * \mathcal{F}^{-1}\left(\prod_{k=1}^{K} f\left(p_k, \omega_{k-1}, \omega_k; \nu\right)\right) = \Gamma(t; \mathcal{P}).
\end{aligned} \tag{3}$$

Therefore, when the poses of the emitter and listener are exchanged, the impulse response becomes:

$$h(t; \mathcal{P}', \omega_l, \omega_e) = G_e(\omega_K; \omega_l) \, \Gamma(t; \mathcal{P}') \, G_l(\omega_0; \omega_e). \tag{4}$$

If the gain patterns of the emitter/listener are omnidirectional (3.2), i.e., $G_e(\cdot) = G_l(\cdot) = 1$, we get:

$$h(t; \mathcal{P}, \omega_e, \omega_l) = h(t; \mathcal{P}', \omega_l, \omega_e). \tag{5}$$

**Impulse Response via Multiple Paths with Reciprocity.** In realistic environments, signal propagation occurs along multiple paths between the emitter and the listener due to reflections, diffractions, and other interactions. Let $\mathcal{P}_n$ denote the $n$-th path. The overall impulse response $h(t)$ is modeled as the superposition of individual path contributions: $h(t) = \sum_{n=1}^{N} h(t; \mathcal{P}_n, \omega_e, \omega_l)$. Since each path-specific response $h(t; \mathcal{P}_n, \omega_e, \omega_l)$ satisfies reciprocity as shown earlier, the sum of these responses also preserves this property. Thus, the full impulse response $h(t)$ remains reciprocal.

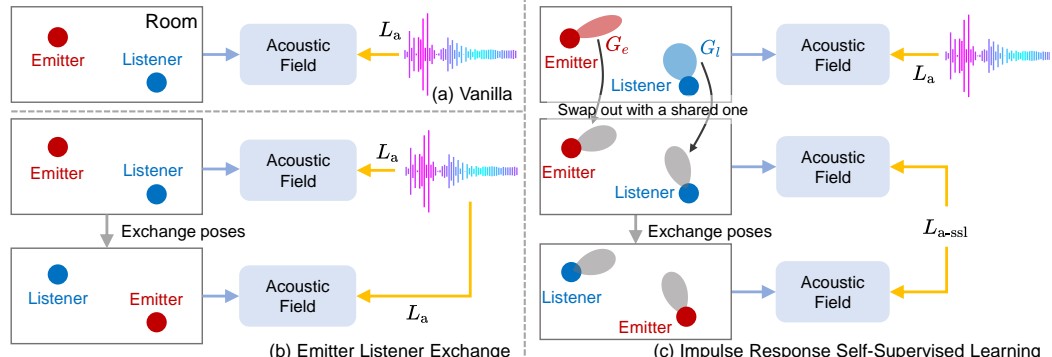

**Figure 3: Leveraging reciprocity for modeling acoustic fields.** a) The vanilla method uses direct supervision with measured impulse responses. b) Versa-ELE enforces response invariance under pose exchange to create physically valid virtual samples. c) Versa-SSL aligns emitter and listener gain pattern to maintain consistency under pose exchange and to enable reciprocity-based self-supervision.

## 3.2 Reciprocity Learning with Versa-ELE

We propose Emitter Listener Exchange (Versa-ELE) to leverage the reciprocity property under omnidirectional emitter/listener gain patterns. We denote one training sample with emitter and listener poses as $(p_e, p_l, \omega_e, \omega_l, h(t))$. We exchange the emitter and listener poses and let the impulse response remain identical, resulting in a physically valid new sample $(p_l, p_e, \omega_l, \omega_e, h(t))$. This is implemented as data augmentation as shown in Fig. 3(b). Versa-ELE leverages the inherent asymmetry in the acoustic field to transform dense listener positions into virtual emitter locations. It effectively mitigates the sparsity of emitter positions by increasing them from a few to $N$, where $N$ is the number of unique listener positions in the dataset. This exchange encourages the model to understand the signal propagation path and improves its performance in resounding tasks (Fig. 4).

Versa-ELE also extends to directional emitters and listeners. Specifically, when both of them have the same gain pattern, i.e., $G_e(\cdot) = G_l(\cdot)$, we have: $G_e(\omega_0; \omega_l) = G_l(\omega_0; \omega_l)$ and $G_e(\omega_K; \omega_e) = G_l(\omega_K; \omega_e)$. These equations ensure that the joint effect of the emitter and listener's pattern on the impulse response remains unchanged after exchanging. With the single-path reciprocity (Eqn. 3), the final impulse responses $h(t)$ remain the same, and Versa-ELE can also be applied here.

## 3.3 Reciprocity Learning with Versa-SSL

Versa-ELE works well if the emitter and listener share the same gain pattern. However, as shown in Fig. 2 right, if they have different patterns, directly exchanging their poses would result in a different impulse response, i.e., $h(t; \mathcal{P}, \omega_e, \omega_l) \neq h(t; \mathcal{P}', \omega_l, \omega_e)$. To deal with this, our solution is to decouple and control the influence of the gain pattern to leverage reciprocity for impulse response learning.

For the rest of this section, we first provide a primer on acoustic volume rendering (AVR) [34], which we find could decouple the influence of the listener's gain pattern. We then introduce a self-supervised learning framework based on this to leverage reciprocity under different gain patterns conditions.

**Primer on AVR and modeling gain patterns.** AVR models the acoustic field with direct control over the listener gain pattern $G_l$, enabled by its acoustic volume rendering technique. To synthesize the impulse response received at a listener location, it integrates signals spherically from all directions:

$$h(t) = \int_\Omega h_\omega(t; p_e, p_l, \omega_e) G_l(\omega; \omega_l) d\omega, \tag{6}$$

where $h_\omega(t; p_e, p_l, \omega_e)$ denotes the predicted signal received by the listener at position $p_l$ from direction $\omega$. $h_\omega(t; p_e, p_l, \omega_e)$ is obtained through acoustic volume rendering, which is similar to the volume rendering in image synthesis, but considers acoustic propagation effects. We ignore the parameters $p_e, p_l, \omega_e$ on the left side ($h(t)$) of the equation for brevity. From this equation, we can manipulate the effect of the listener gain pattern by multiplying weights $G_l(\omega; \omega_l)$ to different receiving directions. It allows us to separately control the listener pattern in the final impulse response synthesis, and we find it to be basis to exploit reciprocity for impulse response training.

**Versa-SSL.** We introduce an impulse response Self-Supervised Learning (Versa-SSL) method that leverages the reciprocity theory by decoupling gain patterns from impulse response modeling. We

model the emitter and listener gain patterns $G_e, G_l$ in the neural acoustic field $\mathbf{F}$ to output impulse response $h(t)$, parameterized by emitter and listener poses:

$$\mathbf{F}(p_e, p_l, \omega_e, \omega_l, G_e, G_l) \rightarrow h(t). \tag{7}$$

We propose to manually replace the emitter/listener pattern with a shared one, i.e. $G$. Under this condition, the acoustic field model should output the same impulse response consistently when emitter/listener poses are exchanged, based on previous discussion (Sec. 3.2):

$$\mathbf{F}(P_l, P_e, \omega_l, \omega_e, G, G) = \mathbf{F}(P_e, P_l, \omega_e, \omega_l, G, G). \tag{8}$$

With this equation defining a reciprocity-based consistency constraint, we formulate a self-supervision learning with pairs created by exchanging poses (Fig. 3(c)). As a result, it forces the neural acoustic field to output consistently after exchanging the emitter/listener poses in a self-supervised way.

Incorporating AVR into our self-supervision framework is not straightforward, as AVR's acoustic field does not explicitly model the emitter pattern, which can not be directly replaced for self-supervision. Instead, we exploit the ability to freely manipulate the listener pattern in Eqn. 6. We propose to keep the emitter's pattern unchanged and swap out the listener's pattern with the emitter's to enable this self-supervision. To achieve this, we first query the neural acoustic field at emitter positions and sample the emitted signals across all directions to get the emitter pattern $G_e$, which is used to swap the listener pattern, making both patterns the same and satisfying the condition for self-supervision.

**Versa-SSL Training.** We use a two-stage training pipeline for Versa-SSL. In the first stage, we let the AVR model to estimate impulse response $h$ and fit into the ground truth $h^*$. This is guided by the audio loss $L_a(h, h^*)$ following AVR. This first stage allows the model to capture the acoustic field of the environment. After it, we extract the emitter pattern from the trained neural acoustic field and use a set of spherical harmonic parameters to encode this pattern, i.e. $G_e$. In the second stage, we swap out the listener pattern with the estimated emitter pattern and force the model to predict identical impulse response pairs ($h_1$ and $h_2$) before and after emitter/listener poses exchange. To enforce this consistency, we use a self-supervision loss $\mathcal{L}_{\text{a-ssl}}(h_1, h_2)$. We use both real sample and self-supervision in the second stage: $\mathcal{L} = \mathcal{L}_a(h, h^*) + \lambda \mathcal{L}_{\text{a-ssl}}(h_1, h_2)$, where $\lambda$ balances two losses. Moreover, since Versa-SSL does not rely on valid data, we can flexibly select any emitter/listener poses. However, excessive flexibility can make the model learn shortcuts. To prevent these, we sample emitter/listener poses from the dataset and gradually add noise to them as Versa-SSL progresses.

**Versa-SSL Inference.** During inference, we can either use the learned listener gain pattern or swap in any head-related transfer functions (HRTFs) for personalized auditory experiences for immersive auditory rendering.

### 3.4 Discussion on the Reciprocity

While the principle of reciprocity holds under idealized conditions [31, 48] (e.g., linear, time-invariant media with symmetric reflection), real-world environments and simulated systems may deviate from these assumptions due to many factors like measurement noise, non-ideal hardware responses, or complex material properties and so on [23]. Rather than assuming perfect reciprocity in the impulse responses, Versa introduces reciprocity as a structural regularization term that enforces consistency between predicted acoustic fields under exchanged emitter–listener configurations. This approach allows the model to benefit from symmetry in wave propagation, even when perfect reciprocity does not strictly hold. We provide some preliminary verification of reciprocity [57] in Sec. 4.2.

## 4 Evaluation

### 4.1 Setup

**Dataset.** We use both simulated and real-world datasets to comprehensively evaluate our methods. We use MeshRIR [29] and RAF [18] as our real datasets, where MeshRIR shows similar emitter/listener gain patterns, and RAF features different emitter/listener patterns. We use the AcoustiX [34] simulator to create synthetic datasets because it provides flexibility to customize emitter and listener gain patterns in the simulation. We apply the AcoustiX simulator on three scenes from iGibson [35] and create two batches of datasets, including using the same emitter/listener gain patterns (AcoustiX-Same) and different gain patterns (AcoustiX-Diff). There are fewer than ten training emitter positions

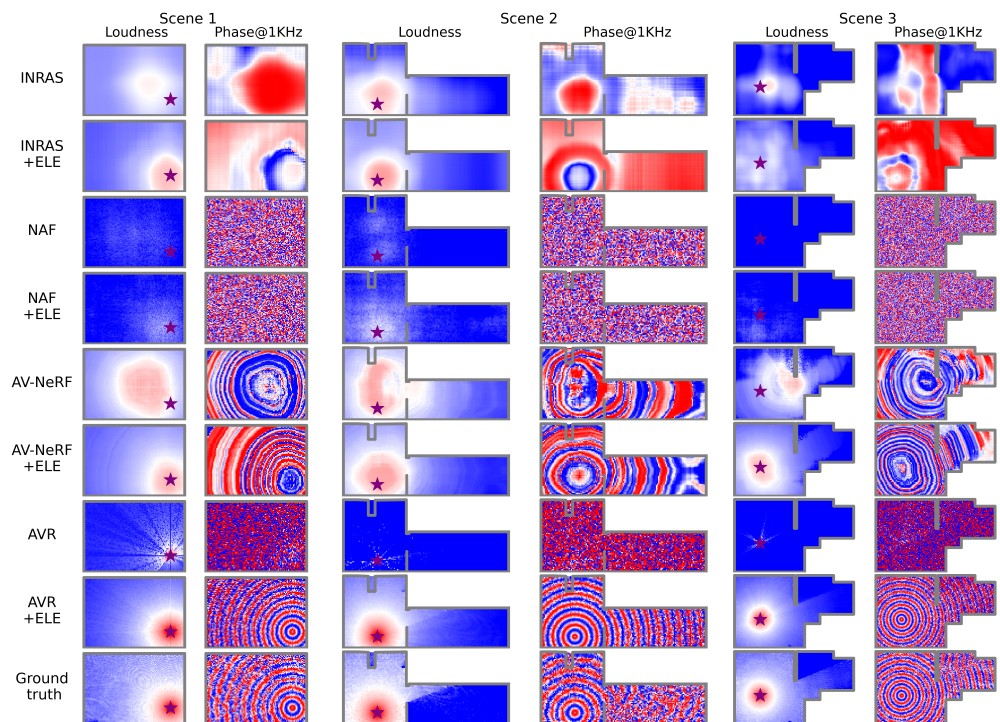

**Figure 4: Comparison of acoustic field predictions across baseline methods with and without Versa-ELE.** We visualize loudness distribution and phase patterns for three simulated scenes (each with *five* training emitter positions and *identical* emitter/listener gain patterns). Purple stars mark emitter positions. Versa-ELE enhances all baseline methods' predictions, particularly in modeling energy distribution around emitters. Ground truth shown in the bottom row.

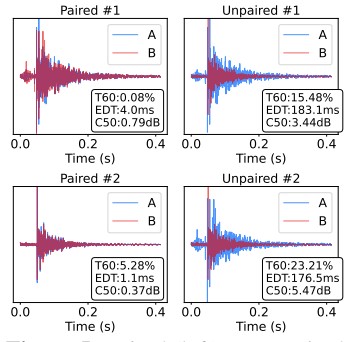

**Figure 5:** Paired (left) vs. unpaired (right) impulse responses.

**Table 1:** Preliminary verification of reciprocity.

| Environment | Variant | Amp | Env | T60 | C50 | EDT |
|---|---|---|---|---|---|---|
| Kitchen | un-paired | 1.74 | 12.0 | 11.3 | 3.69 | 89.1 |
| | paired | **0.24** | **1.8** | **2.1** | **0.29** | **9.8** |
| Conference room | un-paired | 1.09 | 7.5 | 18.7 | 3.35 | 120 |
| | paired | **0.22** | **1.0** | **3.3** | **0.23** | **11.9** |
| Office room | un-paired | 1.54 | 16.5 | 24.5 | 2.49 | 104.5 |
| | paired | **0.23** | **2.0** | **2.2** | **0.18** | **11.0** |
| Simulation | un-paired | 0.43 | 5.2 | 16.6 | 1.95 | 98.5 |
| | paired 10k rays | 0.13 | 0.14 | 5.9 | 0.4 | 7.1 |
| | paired 100k rays | 0.07 | 0.07 | 2.1 | 0.12 | 3.8 |
| | paired 1000k rays | **0.06** | **0.03** | **0.48** | **0.05** | **0.8** |

in each room for all simulated datasets, with densely sampled emitter positions for evaluation. Please refer to Appx. A.1 for more details.

**Models.** We compare the performance of our method with traditional audio encoding baselines using linear and nearest-neighbor interpolation [39, 54, 59]. We implement various neural acoustic field models including NAF [39], INRAS [54], AV-NeRF [37], and AVR [34]. We apply Versa-ELE (Sec. 3.2) to all these model. We only apply Versa-SSL (Sec. 3.3) to AVR as it is the only one to explicitly model the gain patterns. In addition, we also implement DiffRIR [59] as a baseline. However, since DiffRIR explicitly traces acoustic paths, Versa is excluded from it because the paths naturally remain consistent after exchanging emitter and listener positions. Please refer to Appx. A.3 for implementation details.

**Evaluation Metrics.** Following prior work[54, 37, 34], we measure the energy trend of impulse response by Reverberation Time (T60), Clarity (C50), and Early Decay Time (EDT). To evaluate the waveform correctness, we include envelope error (Env.), frequency domain amplitude (Amp.) error, and multi-resolution short-time Fourier transform error (STFT).

| Method | Scene 1 | | | | | | Scene 2 | | | | | | Scene 3 | | | | | |
|---|---|---|---|---|---|---|---|---|---|---|---|---|---|---|---|---|---|---|
| | STFT | Amp. | Env. | T60 | C50 | EDT | STFT | Amp. | Env. | T60 | C50 | EDT | STFT | Amp. | Env. | T60 | C50 | EDT |
| NN | 2.87 | 0.35 | 3.3 | 15.8 | 2.84 | 19.6 | 3.54 | 0.47 | 3.1 | 43.8 | 10.71 | 25.5 | 3.29 | 0.47 | 3.9 | 48.9 | 7.42 | 35.6 |
| Linear | 2.72 | 0.94 | 6.0 | 15.9 | 3.37 | 20.8 | 3.32 | 1.09 | 5.6 | 69.7 | 15.64 | 37.3 | 3.15 | 1.08 | 6.7 | 62.8 | 11.3 | 51.2 |
| DiffRIR | 1.57 | 0.36 | 8.0 | 12.2 | 2.68 | 23.9 | 1.63 | 0.48 | 6.7 | 13.6 | 2.57 | 19.4 | 1.74 | 0.76 | 8.8 | 31.1 | 5.17 | 42.5 |
| INRAS | 1.96 | 0.72 | 3.6 | 12.3 | 2.71 | 24.6 | 1.96 | 0.72 | 3.6 | 12.3 | 2.71 | 24.6 | 4.22 | 1.03 | 3.5 | 93.5 | 7.14 | 50.3 |
| w/ ELE | **1.36** | **0.23** | **2.8** | **12.0** | **1.72** | **13.6** | **1.81** | **0.31** | **2.6** | **11.6** | **1.98** | **17.7** | **1.67** | **0.41** | **3.4** | **20.1** | **2.79** | **22.7** |
| NAF | 4.69 | 0.69 | 3.7 | 14.1 | 2.73 | 23.3 | 7.29 | 0.76 | 3.5 | 16.9 | 5.47 | 27.3 | 5.61 | 0.74 | 3.4 | 28.2 | 5.86 | 43.9 |
| w/ ELE | **3.27** | **0.66** | **3.5** | **13.2** | **2.2** | **18.2** | **3.27** | **0.66** | **3.1** | **14.1** | **2.17** | **18.2** | **3.37** | **0.58** | **3.3** | **23.7** | **3.31** | **26.5** |
| AV-NeRF | 1.69 | 0.85 | 4.6 | 11.0 | 4.98 | 44.1 | 1.87 | 2.35 | 4.8 | 18.7 | 5.26 | 36.6 | 1.85 | 1.17 | 5.2 | 26.2 | 6.09 | 49.3 |
| w/ ELE | **1.15** | **0.16** | **2.4** | **10.7** | **1.70** | **12.5** | **1.41** | **0.35** | **2.9** | **12.5** | **2.43** | **20.3** | **1.36** | **0.42** | **3.5** | **19.8** | **2.97** | **23.7** |
| AVR | 1.82 | 0.40 | 3.1 | 14.3 | 2.09 | 15.9 | 2.56 | 0.46 | 2.9 | 20.2 | 3.43 | 37.6 | 2.65 | 0.69 | 3.6 | 28.6 | 2.96 | 42.3 |
| w/ ELE | **1.12** | **0.15** | **2.20** | **10.5** | **1.61** | **12.2** | **1.28** | **0.19** | **1.7** | **11.2** | **1.31** | **12.9** | **1.21** | **0.32** | **2.3** | **18.9** | **2.33** | **20.3** |

**Table 2:** Quantitative results on AcoustiX-Same *(5 emitter positions)* when emitter/listener shares a gain pattern. We deploy Versa-ELE (i.e., w/ ELE) on existing neural acoustic fields.

## 4.2 Results

**Preliminary Verification of Reciprocity.** We begin by empirically verifying the acoustic reciprocity property. We collect impulse responses in an office room, a conference room, and a kitchen. Recordings are performed using an *omnidirectional microphone* [4] and an *omnidirectional loudspeaker* [10], both connected via an audio interface [7] to ensure synchronization. We capture impulse responses between pairs of locations (A, B) within each environment. For every pair, we measure the impulse response emitted from A to B and the reciprocal one from B to A. According to our analysis, these two impulse responses should be identical since both the microphone and speaker are omnidirectional. To quantify reciprocity, we collect 20 such pairs in each scene and compute the distance between (i) paired impulse responses with swapped emitter/listener positions and (ii) un-paired ones for rest of the data. As shown in Tab. 1, the paired ones exhibit ≈10× smaller acoustic distance compared with the un-paired ones in all real-world scenes. Examples of paired and un-paired impulse responses are presented in Fig. 5, the waveform similarity between the paired one also confirm that reciprocity is well preserved. We also replicate this experiment in the AcoustiX simulated dataset. We vary the number of rays in the simulation to analyze how rendering granularity affects reciprocity consistency. In simulated dataset, paired ones achieve ≈100× smaller distances with 1000k rays.

**Results of Versa-ELE.** We first evaluate Versa-ELE on similar emitter/listener gain patterns datasets, namely the AcoustiX-Same dataset in Tab. 2 and the real-world MeshRIR dataset in Tab. 3. We find that ELE can be applied to all existing learning-based impulse response estimation methods and improve their performance by a large margin.

Fig. 4 shows a qualitative visualization of the spatial field distributions in birds-eye view. With an unseen fixed emitter, we densely sample listener locations in the entire scene and plot the loudness and phase map across three scenes. With Versa-ELE, all baseline methods (INRAS, NAF, AV-NeRF, and AVR) achieve notably enhanced spatial field distributions that better reflect the emitter positions. Notably, the improvement on the AVR model is the most significant. It produces a field that best aligns with the ground truth in both loudness and phase map.

| Method | STFT | Amp. | Env. | T60 | C50 | EDT |
|---|---|---|---|---|---|---|
| NN | 2.46 | 0.42 | 1.33 | 7.6 | 2.34 | 26.9 |
| Linear | 2.65 | 1.22 | 2.76 | 9.9 | 2.76 | 30.9 |
| INRAS | 1.82 | 0.54 | 1.32 | 13.6 | 4.04 | 47.8 |
| w/ ELE | **1.73** | **0.36** | **1.22** | **12.9** | **2.51** | **28.2** |
| NAF | 4.81 | 0.69 | 1.18 | 9.9 | 2.28 | 26.2 |
| w/ ELE | **4.43** | **0.65** | **1.14** | **8.6** | **2.17** | **23.5** |
| AVR | 3.78 | 0.61 | 1.20 | 15.6 | 4.97 | 65.2 |
| w/ ELE | **1.42** | **0.33** | **1.02** | **7.2** | **1.79** | **20.9** |

**Table 3:** Evaluation on Versa-ELE method on MeshRIR *(7 emitter positions)*.

Though Versa-ELE does not consider different emitter/listener gain patterns, we evaluate it on such datasets including AcoustiX-Diff in Tab. 4 and RAF-Furnished in Tab. 5. Results show that Versa-ELE can still improve these baselines by a reasonable margin, showing ELE's wide applicability to various datasets and real-world settings, especially on neural acoustic fields that cannot incorporate gain patterns modeling (NAF, INRAS, AV-NeRF). Although Versa-ELE may not accurately account for varying gain patterns, it enhances the model's awareness of reciprocity in the propagation path, leading to improved acoustic field modeling. Quantitatively, averaging over all models and simulated scenes, Versa-ELE improves their performance by 34% at C50, and 31% at STFT.

**Results of Versa-SSL.** We evaluate Versa-SSL on datasets with different emitter/listener gain patterns, including the AcoustiX-Diff dataset in Tab. 4 and the RAF dataset in Tab. 5. While we show that Versa-ELE can be applied to all the existing neural network-based impulse response estimation methods and

| Method | Scene 1 | | | | | | Scene 2 | | | | | | Scene 3 | | | | | |
|---|---|---|---|---|---|---|---|---|---|---|---|---|---|---|---|---|---|---|
| | STFT | Amp. | Env. | T60 | C50 | EDT | STFT | Amp. | Env. | T60 | C50 | EDT | STFT | Amp. | Env. | T60 | C50 | EDT |
| NN | 2.86 | 0.48 | 5.8 | 69.1 | 13.9 | 104.4 | 2.52 | 0.48 | 4.9 | 77.7 | 18.1 | 116.5 | 2.74 | 0.60 | 9.5 | 60.9 | 17.13 | 208.5 |
| Linear | 2.44 | 0.67 | 9.1 | 70.5 | 18.1 | 233.4 | 1.96 | 0.95 | 8.1 | 77.5 | 22.2 | 238.8 | 2.26 | 0.69 | 11.6 | 61.1 | 17.34 | 216.9 |
| DiffRIR | 1.59 | 0.35 | 7.3 | 16.5 | 2.52 | 24.2 | 1.61 | 0.68 | 7.4 | 18.8 | 3.17 | 24.5 | 1.68 | 0.64 | 8.2 | 26.7 | 3.91 | 32.8 |
| INRAS | 3.12 | 1.05 | 5.3 | 30.9 | 6.45 | 48.8 | 2.65 | 1.56 | 3.8 | 67.9 | 6.71 | 44.3 | 4.12 | 1.05 | 4.3 | 94.8 | 6.94 | 67.1 |
| w/ ELE | **1.91** | **0.31** | **4.3** | **16.8** | **2.93** | **23.1** | **1.76** | **0.64** | **3.4** | **25.5** | **4.36** | **31.3** | **1.58** | **0.38** | **3.9** | **38.4** | **3.13** | **29.6** |
| NAF | 5.21 | 0.59 | 4.8 | 37.5 | 4.17 | 34.9 | 6.21 | 0.73 | 3.2 | 40.5 | 5.61 | 40.3 | 8.12 | 0.71 | 4.2 | 57.7 | 6.41 | 51.6 |
| w/ ELE | **4.12** | **0.56** | **4.5** | **20.0** | **2.97** | **24.4** | **4.51** | **0.61** | **3.1** | **22.8** | **4.43** | **32.5** | **4.09** | **0.56** | **4.1** | **32.4** | **4.50** | **35.9** |
| AV-NeRF | 1.99 | 1.17 | 6.3 | 18.5 | 4.32 | 43.1 | 2.16 | 1.22 | 3.6 | **16.5** | 3.01 | 24.7 | 2.27 | 1.49 | 6.0 | 35.1 | 5.45 | 49.7 |
| w/ ELE | **1.57** | **0.35** | **4.7** | **16.1** | **2.72** | **21.6** | **1.61** | **0.42** | **3.0** | 16.9 | **2.85** | **22.9** | **1.62** | **0.72** | **4.7** | **32.7** | **4.97** | **40.1** |
| AVR | 2.65 | 0.77 | 4.1 | 36.1 | 4.96 | 41.7 | 2.65 | 0.95 | 3.6 | 76.1 | 5.50 | 64.3 | 2.45 | 0.95 | 4.2 | 46.2 | 5.69 | 51.2 |
| w/ ELE | 1.83 | 0.34 | 3.9 | 17.3 | 2.83 | 29.2 | 1.79 | 0.48 | 2.5 | 24.4 | 3.79 | 26.8 | 1.62 | 0.48 | 3.8 | 32.1 | 4.36 | 42.5 |
| w/ SSL | **1.39** | **0.25** | **3.6** | **15.7** | **2.09** | **21.5** | **1.26** | **0.27** | **2.3** | **15.4** | **1.31** | **21.6** | **1.31** | **0.27** | **3.5** | **19.4** | **2.07** | **24.1** |

**Table 4:** Results on AcoustiX-Diff *(5 emitter positions)* when emitter and listener have different gain patterns. We apply Versa-ELE for all the models, and Versa-SSL method on AVR.

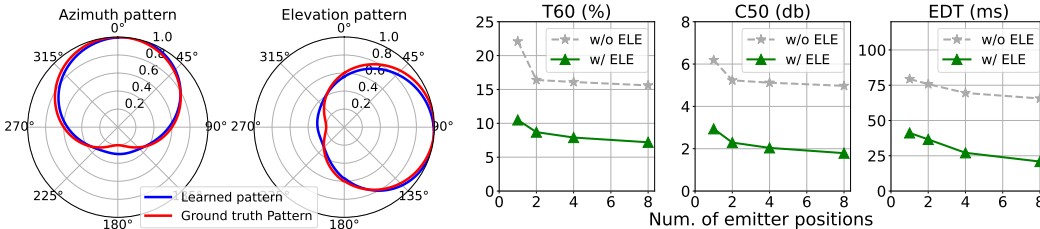

**Figure 6:** Learned gain pattern with direction-integrated impulse responses.

**Figure 7:** Performance across different numbers of emitter positions.

improve their performance, Versa-SSL could further surpass Versa-ELE by 24% at C50, and 48% at STFT on the AVR model. In total, Versa-SSL improves vanilla AVR by 49% at C50, and 66% at STFT.

These improvements can be largely attributed to handling the gain pattern correctly with self-supervision, which makes the model aware of reciprocity in the acoustic propagation.

Recall Fig. 1, we visualize the spatial field distributions when the emitter and listener have different gain patterns. With both the ELE and SSL strategies, the AVR model can produce more accurate phase maps that align with the ground-truth. However, Versa-ELE cannot accurately reflect the direction of the emitter in the loudness map, as we force the model to learn inaccurate impulse responses after exchanging poses, which causes directionality confusion. In contrast, as Versa-SSL deals with gain pattern

| Method | Amp. | Env. | T60 | C50 | EDT | STFT |
|---|---|---|---|---|---|---|
| NN | 0.75 | 11.8 | 23.4 | 6.5 | 167.1 | 4.74 |
| Linear | 0.99 | 16.9 | 29.1 | 7.6 | 201.9 | 4.28 |
| DiffRIR | 0.71 | 22.3 | 28.9 | 4.23 | 154.3 | 1.61 |
| INRAS | 0.66 | 8.8 | 19.4 | 5.63 | 149.6 | **3.21** |
| w/ ELE | **0.50** | **6.6** | **16.3** | **4.67** | **117.3** | 3.43 |
| NAF | 0.65 | 6.0 | 15.4 | 5.91 | 123.6 | 6.22 |
| w/ ELE | **0.63** | **5.9** | **14.7** | **5.20** | **109.5** | **5.46** |
| AV-NeRF | 1.27 | 10.1 | 15.3 | 4.57 | 110.7 | 2.55 |
| w/ ELE | **0.66** | **6.5** | **12.7** | **4.12** | **96.4** | **2.41** |
| AVR | 0.59 | 5.8 | 14.8 | 4.24 | 99.6 | 2.35 |
| w/ ELE | 0.51 | 5.2 | 13.9 | 4.14 | 95.8 | 2.16 |
| w/ SSL | **0.48** | **4.9** | **12.2** | **3.65** | **83.6** | **1.91** |

**Table 5:** Results on RAF-Furnished (*4 emitters*).

properly through self-supervision, it helps to build a correct acoustic field with better directionality. We also show our estimated gain pattern on the AcoustiX-Diff in Fig. 6, indicating that our estimation is close to the ground truth.

**Perceptual user study.** We conducted a user study to evaluate the subjective quality of impulse response estimation. In total, we generate eight audio clips from three simulated scenes involving moving listeners and speakers. We also simulate reference impulse responses as ground truth. Each estimated and reference impulse response is convolved with dry sound, and participants watch a video from the microphone's perspective for added real-

| | ELE vs Vanilla | SSL vs ELE |
|---|---|---|
| Volume | 90.0% | 88.3% |
| Directional | 90.0% | 90.0% |
| Overall | 93.3% | 85.0% |

**Table 6:** Results of the perceptual user study.

ism. They are then asked to compare the generated audio to the reference in terms of volume, directional cues, and overall similarity.

Our user study with 15 participants shows that 93.3% of responses preferred Versa-ELE over the vanilla model in terms of overall similarity. Additionally, 85% of responses indicated that Versa-SSL

outperforms Versa-ELE in sound volume and directional cues. As shown in Tab. 6. Please find more details in the Appx. B.3.

### 4.3 Ablation studies

*What is the influence of number of emitter positions?* We analyze how varying the number of emitter positions affects Versa. As shown in Fig. 7, while increasing the number of emitter positions in the dataset can improve the vanilla AVR performance, with the help of Versa-ELE, we can further improve the model's performance by 50%. This shows that Versa consistently improves the vanilla model by a large margin across different numbers of emitter positions. More ablation results on different methods with various numbers of emitter and listener positions can be found in Appx. B.1.

| Study objective | Variant | Amp | Env | T60 | C50 | EDT |
|---|---|---|---|---|---|---|
| *Does the model learn reciprocity?* | NAF w/o Versa-ELE | 0.33 | 5.9 | 13.5 | 2.83 | 50.4 |
| | NAF w/ Versa-ELE | **0.27** | **3.0** | **5.4** | **0.45** | **8.4** |
| | AVR w/o Versa-ELE | 0.54 | 10.6 | 16.6 | 4.13 | 86.9 |
| | AVR w/ Versa-ELE | **0.06** | **0.5** | **4.8** | **0.61** | **10.8** |
| *Comparison with common SSL loss* | Vanilla | 0.89 | 3.97 | 52.8 | 5.35 | 52.4 |
| | Contrastive loss | 0.47 | 3.45 | 25.3 | 3.22 | 36.0 |
| | Versa-SSL | **0.26** | **3.10** | **16.8** | **1.82** | **22.4** |

**Table 7:** Ablation on the reciprocity verification on the trained model and self-supervision loss.

*Does the model learn reciprocity with Versa?* We emphasize the importance of verifying whether the neural acoustic field models trained with Versa indeed exhibit reciprocity. To this end, we randomly query the trained AVR and NAF models with swapped emitter–listener pairs across different scenes and measure the similarity between the predicted impulse responses. As summarized in Tab. 7, the models trained with Versa produce paired predictions that achieve substantially lower acoustic metric differences compared to those trained without Versa. These results demonstrate that Versa effectively encourages the neural acoustic field to learn and preserve reciprocity.

*How effective is Versa-SSL compared with standard contrastive loss?* We also evaluate standard contrastive loss by treating emitter/listener swapped pairs as positives and others as negatives. As shown in the bottom of Tab. 7, contrastive loss yields smaller gains over the Versa-SSL. We attribute this to the ambiguous notion of negatives pairs in acoustic field modeling. Unlike vision tasks with clearly separable categories, impulse responses from different spatial locations can exhibit nearly identical patterns, while responses from nearby poses may differ drastically due to occlusions or boundary effects. As a result, spatial distance alone does not reliably define negative pairs.

## 5 Discussion

**Conclusion**. In this work, we leverage acoustic reciprocity to improve impulse response estimation, addressing the challenge of resounding under sparse emitter configurations. We introduce the Versa-ELE to augment training data which leads to substantial improvements in neural acoustic models. In addition, we propose a Versa-SSL to handle complex gain pattern asymmetries, enabling more accurate acoustic field synthesis. More broadly, our results highlight a paradigm where fundamental physical laws can be translated into machine learning training strategies, improving generalization under sparse data. This principle applies beyond acoustics to light transport and RF propagation, where physical symmetries also govern wave interactions. By embedding these constraints directly into the learning process, models can extrapolate more robustly to unseen configurations and maintain physical correctness even under limited supervision. We envision that such physics-grounded learning frameworks will serve as a foundation for future research at the intersection of simulation, perception and more across multiple sensing modalities.

**Limitation and Future Works.** Despite its effectiveness, Versa still requires a non-sparse number of listener positions per scene, limiting its scalability in scenarios where dense microphone deployment is impractical. Moreover, our current acoustic modeling relies on ray-based geometric acoustics, which may be less accurate for low-frequency sounds whose wavelengths are comparable to or larger than scene dimensions. Future work can explore few-shot or zero-shot approaches by integrating more audio-visual correspondence and other acoustic properties beyond reciprocity, which can further reduce the need for dense sampling of listener positions.

## Acknowledgments

We thank the members of the WAVES Lab at the University of Pennsylvania for their valuable feedback. We are grateful to the anonymous reviewers for their insightful comments and suggestions.

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

# A  Additional Implementation Details

## A.1  Dataset Details

**AcoustiX.** We show the emitter locations in three different simulated scenes from the iGibson dataset [35] in Fig. 8. For each scene, there are five emitter positions in the training set, and each with around 5K emitter pairs, resulting 25K training samples. We randomly sample 10 novel emitter locations to construct the testing set, resulting in a total of 15K samples.

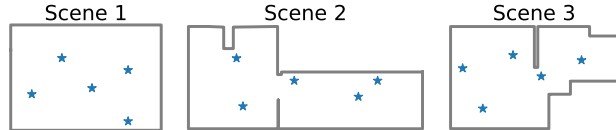

**Figure 8:** Visualization of dataset emitter placements.

**MeshRIR.** We subdivide the S32-441 variant to use 7 emitter positions (3K emitter/listener pairs) as training samples, while reserving 25 emitter positions (11K total emitter/listener pairs) for testing.

**RAF.** We randomly select 4 emitter positions in both FurnishedRoom and EmptyRoom split, resulting in a total of 1.5K training samples, and we resample 6K testing samples in each setting.

**Dataset setup.** For similar emitter/listener gain patterns, we utilize the S32-M441 split MeshRIR dataset and sub-divide it to include seven emitter positions in the training set. We create simulated AcoustiX datasets while maintaining the emitter and listener to have the same pattern, naming it AcoustiX-Same. For different emitter/listener gain patterns, we use real-world RAF Furnishedroom and Emptyroom split, while re-dividing these to include several sparse ($< 10$) emitter positions for training. We again use AcoustiX to simulate the same three scenes but set different emitter and listener gain patterns, naming it AcoustiX-Diff. The impulse responses are resampled to 16 kHz sampling rate for the AcoustiX and RAF datasets and 24 kHz sampling rate for the MeshRIR dataset.

## A.2  Model Details

**NAF.** We create 3D grid features for original NAF [39] implementation. We use a STFT window size of 256 (hop size 64) to deal with 16 KHz impulse response and a STFT window size of 512 (hop size 128) to deal with 24 KHz impulse response. We let the model directly predict the amplitude of the spectrogram and use a random phase to synthesize the final impulse response. We only implement the Versa-ELE method on NAF. We fully exchange the emitter and listener poses and generate a new impulse response sample in the AcoustiX and MeshRIR datasets. For the RAF dataset, we only exchange the emitter and listener positions, since no listener orientation is provided.

**AV-NeRF.** We use AV-NeRF [37] SoundSpace variant to predict the room impulse response on all the experiments. We train a vision NeRF model first and reconstruct novel-view RGB and depth images at the listener location and direction in 256×256 resolution. We used a frozen ResNet-18 trained on ImageNet as the feature extractor. We follow a similar implementation for the Versa strategy as in NAF.

**INRAS.** We use the 3d positions of emitter, listener, and bounce points in the scene and the orientation of emitter and listener as network input, similar in [18]. We randomly sample 1024 points in the scene to represent the scene geometry. We also follow a similar implementation for the Versa strategy as in NAF.

**AVR.** We implement the Versa-ELE method on AVR [34] by fully exchanging the emitter and listener poses and generating a new impulse response sample. For Versa-SSL, to estimate the emitter gain pattern in AVR, we query the network with the emitter's positions to obtain the signals transmitted from these positions in various directions. To reconstruct the gain pattern, the directionality is characterized by the average signal energy in each specific direction, which is subsequently normalized across all directions.

**DiffRIR.** DiffRIR [59] takes the emitter locations and listener locations to trace paths between them. Then it optimizes a set of acoustic parameters along the traced path. For each scene, we use around 40 simplified meshes to represent the room geometries. We used a maximum of 3-bounce ray tracing

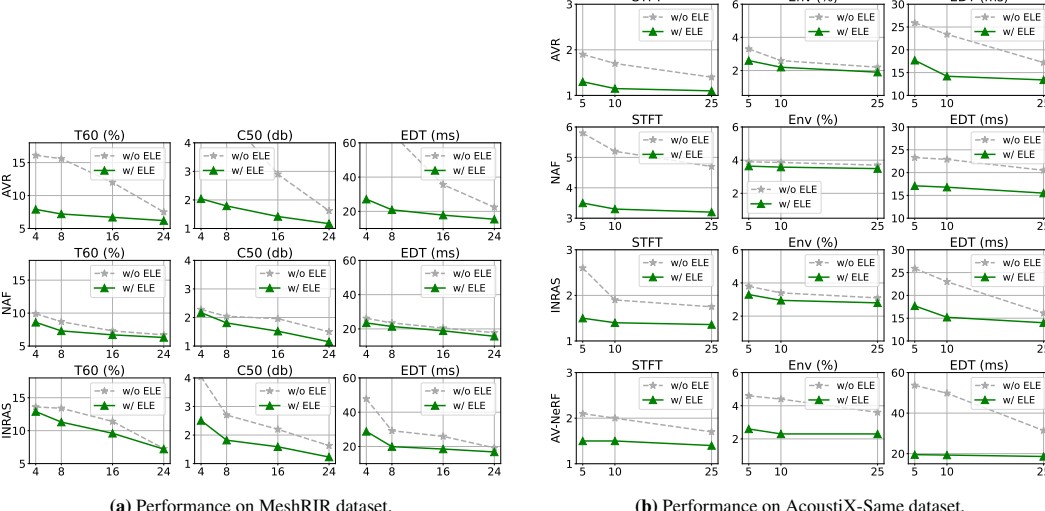

**(a)** Performance on MeshRIR dataset.        **(b)** Performance on AcoustiX-Same dataset.

**Figure 9:** Performance comparison of different methods on (a) MeshRIR and (b) AcoustiX-Same datasets.

with up to 10 axial bounces to keep the training time manageable. DiffRIR is excluded from using any of our Versa method because it already traces the propagation paths, enforcing that the reciprocity property holds.

### A.3 Implementation Details

For NAF, INRAS, AV-NeRF, and AVR experiments, we use the AdamW optimizer [28] with a cosine learning rate scheduler that starts from $10^{-3}$ and decays to $10^{-4}$, with a batch size of 64 for NAF, INRAS, and AV-NeRF, and a batch size of 4 for AVR. To implement Versa-SSL method on AVR, we use a fourth-order spherical harmonics [27] to estimate the emitter gain pattern. We initiate the second stage of Versa-SSL on AVR halfway through the total training epochs, gradually adding position variation to a standard deviation of 0.3 m. We use a loss weight $\lambda = 0.8$. In the DiffRIR experiments, we use an AdamW optimizer with a learning rate $10^{-3}$ and a batch size of 4. For AV-NeRF, we use nerfacto [55] to render RGB and depth images following [37, 18]. We train all the models on NVIDIA L40s GPUs for 200 epochs.

## B Additional Experiment Results

### B.1 Quantitative Results

**More results on RAF dataset**. We show more results on the RAF-EmptyRoom on Tab. 8, our evaluation metrics show the effectiveness for both Versa-SSL and Versa-ELE as well.

**More results on different scales of the dataset.** To further show that our proposed method is robust to different methods with various numbers of emitter positions, we plot the metrics for each method in Fig. 9a and 9b on MeshRIR and AcoustiX-Same dataset. Versa can consistently improve each baseline model under different numbers of emitter settings, even with denser emitter placements.

In addition to emitter positions, we also show that our method is effective when training with fewer listeners. We experiment on AcoustiX, which only uses 10x less (around 400) listener positions per emitter. Our method is also effective in this case with fewer listeners and improves baseline performance as shown in the

| Method | Amp. | Env. | T60 | C50 | EDT | STFT |
|---|---|---|---|---|---|---|
| NN | 1.95 | 11.1 | 15.9 | 7.25 | 174.4 | 5.42 |
| Linear | 2.63 | 12.9 | 27.7 | 8.03 | 216.4 | 4.97 |
| DiffRIR | 1.40 | 4.2 | 24.4 | 3.29 | 122.0 | 1.93 |
| INRAS | 1.09 | 16.2 | 19.1 | 7.13 | 123.2 | 2.59 |
| w/ ELE | **1.07** | **9.6** | **18.2** | **4.25** | **89.2** | **2.54** |
| NAF | 0.99 | 7.3 | **13.5** | **3.87** | 84.7 | 3.42 |
| w/ ELE | **0.89** | **6.6** | **13.5** | 4.12 | **84.1** | **3.39** |
| AV-NeRF | 2.68 | 11.9 | **11.9** | 3.75 | 88.8 | 2.44 |
| w/ ELE | **1.14** | **7.49** | 12.2 | **3.45** | **80.8** | **2.21** |
| AVR | 1.21 | 9.8 | 12.8 | 4.48 | 109.5 | 2.98 |
| w/ ELE | 0.98 | 7.8 | 12.4 | 3.95 | 85.6 | 2.43 |
| w/ SSL | **0.78** | **6.2** | **11.0** | **3.21** | **71.3** | **1.83** |

**Table 8:** Results on RAF-Furnished (*4 emitters*).

| Variant | STFT | Amp. | Env. | T60 | C50 | EDT |
|---|---|---|---|---|---|---|
| INRAS (5k) | 1.9 | 0.72 | 3.6 | 12.3 | 2.7 | 24.6 |
| 10x less | 2.6 | 1.72 | 3.8 | 19.5 | 3.4 | 25.9 |
| 10x less w/ ELE | **1.5** | **0.43** | **3.3** | 16.5 | **2.6** | **17.7** |
| NAF (5k) | 4.7 | 0.69 | 3.7 | 14.1 | 2.7 | 23.3 |
| 10x less | 5.8 | 0.72 | 3.9 | 17.6 | 2.9 | 25.8 |
| 10x less w/ ELE | **3.5** | **0.67** | **3.6** | 16.5 | **2.5** | **18.3** |
| AV-NeRF (5k) | 1.7 | 0.85 | 4.6 | 11.0 | 5.0 | 44.5 |
| 10x less | 2.1 | 1.33 | 4.6 | 17.5 | 6.2 | 53.7 |
| 10x less w/ ELE | **1.5** | **0.32** | **2.6** | 16.8 | **3.8** | **19.5** |
| AVR (5k) | 1.8 | 0.4 | 3.1 | 14.3 | 2.1 | 15.9 |
| 10x less | 1.9 | 0.55 | 3.3 | 15.1 | 2.6 | 16.9 |
| 10x less w/ ELE | **1.3** | **0.24** | **2.6** | **12.6** | **2.0** | **14.3** |

**(a)** Versa is also effective with 10× fewer listeners.

| | STFT | Amp. | Env. | T60 | C50 | EDT |
|---|---|---|---|---|---|---|
| NN | 2.6 | 0.83 | 11.4 | 26.1 | 4.4 | 52.8 |
| Linear | 2.5 | 0.84 | 13.4 | 23.6 | 4.2 | 47.7 |
| NAF | 3.3 | 0.79 | 10.4 | 23.7 | 3.6 | 49.9 |
| w/ ELE | **2.6** | **0.54** | **7.9** | **17.9** | **2.3** | **28.2** |
| INRAS | 2.5 | 0.98 | 9.8 | 26.5 | 3.8 | 47.7 |
| w/ ELE | **2.2** | **0.76** | **7.5** | **18.9** | **2.3** | **31.6** |
| AVR | 2.2 | 0.82 | 11.1 | 22.9 | 2.9 | 45.4 |
| w/ ELE | **1.7** | **0.57** | **7.3** | **16.8** | **2.1** | **27.9** |

**(b)** Results on GWA dataset.

**Table 9:** Quantitative comparison across datasets. (a) Fewer listener positions; (b) GWA dataset.

Tab 9a. With Versa, each model can predict much better results when having 10X fewer listener samples, even compared with the vanilla method with 10X more samples.

**More results on GWA dataset.** Furthermore, we also experiment on the GWA dataset [56] that simulates impulse response with a hybrid ray-based and wave-based method. Each room contains only 5 emitter positions, and there are 100 listener positions per emitter. We demonstrate that Versa still consistently improves the performance as shown in Tab. 9b.

## B.2 Discussion on the permutation invariant structure.

Versa is grounded in the physical principle of acoustic reciprocity. Based on the reciprocity principle, it is possible to design permutation-invariant neural networks like shared encoders for emitter and listener. Such architecture enforces symmetry unconditionally, regardless of whether impulse response remains the same with direct swapping. However, emitters and listeners can have different directional gain patterns, which breaks the impulse response invariance. Enforcing strict symmetry in such cases would mislead the model to make incorrect assumptions.

We provide results on AcoustiX-Diff with a permutation-invariant version of NAF. As shown in the Tab. 10, this structure can improve the performance on the original NAF under the sparse emitters settings. As this design treats the emitter/listener exactly the same, it can alleviate the problem of sparse emitters. However, the permutation-invariant design strictly forces the model to output symmetric results and does not consider the unique features of the emitter or listener including different gain patterns. In contrast, vanilla neural acoustic models that encode emitter/listener features separately are not strictly constrained to this inappropriate symmetry and can benefit from the Versa-ELE for better performance. Finally, AVR with Versa-SSL achieves the best overall performance as it correctly handles the different gain patterns.

| Variant | STFT | Amp. | Env. | T60 | C50 | EDT |
|---|---|---|---|---|---|---|
| NAF | 6.51 | 0.68 | 4.1 | 45.6 | 5.49 | 42.3 |
| NAF + permutation | 4.47 | 0.62 | 4.0 | 25.9 | 3.90 | 37.9 |
| NAF + Versa-ELE | 4.24 | 0.58 | 3.9 | 15.1 | 3.67 | 30.9 |
| AVR + Versa-SSL | **1.32** | **0.26** | **3.1** | 16.8 | **1.85** | **22.4** |

**Table 10:** Comparison on the permutation invariant structure.

## B.3 Perceptual Study

**Study setup.** We render eight audio-video clips using AVR and various Versa variants. Each clip includes three impulse responses: a reference (simulated with AcoustiX), a baseline (either vanilla or Versa-ELE), and an improved method (either Versa-ELE or Versa-SSL). This setup allows us to compare the vanilla method versus Versa-ELE and Versa-ELE versus Versa-SSL. We convolve each

impulse response with dry sounds and pair it with microphone-view videos and trajectory animations. Participants first watch and listen to the reference clip, then evaluate the baseline and improved clips on three criteria:

- Volume accuracy: Which clip conveys changes in volume more accurately?
- Directional accuracy: Which audio made it easier for you to identify the direction of the sound source more accurately?
- Overall similarity: Which clip aligns better with the reference overall?

We provide a snapshot of our user study interfaces in Fig. 10.

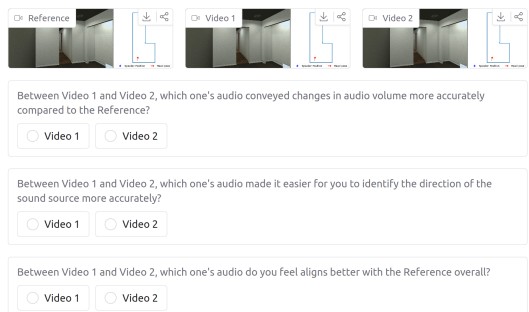

**Figure 10:** The interface for the user study evaluations.

**Result.** We present the user study results in Tab. 6, averaging participants' choices across all audio clips (in total 15 participants). 90% of responses indicate that Versa-ELE better aligns with the reference impulse response in terms of volume, directional accuracy, and overall similarity. Additionally, over 85% of responses favor Versa-SSL over Versa-ELE as well. With these user study results, we demonstrate that our method can greatly improve the spatial audio listening experience, including sound direction and loudness.

**Perceptual metric.** We computed the Deep Perceptual Audio Metric (DPAM) on the audio clips rendered for user study. As shown in Tab. 11, Versa-ELE reduces DPAM error by 20%, and Versa-SSL further reduces error by 36% compared with Versa-ELE on the AVR method. Versa-ELE also boosts the performance of other methods by 18%. These show our method can improve the perceptual similarity to ground truth. These results are consistent with the user study findings and reinforce the perceptual benefit of our method in the resounding task.

| Method | DPAM score |
|---|---|
| NAF + Versa-ELE | 1.53 -> 1.26 |
| INRAS + Versa-ELE | 1.42 -> 1.19 |
| AV-NeRF + Versa-ELE | 1.47 -> 1.15 |
| AVR + Versa-ELE | 1.44 -> 1.17 |
| AVR + Versa-SSL | 1.44 -> 0.86 |

**Table 11:** Results of the perceptual metric.

## B.4 Details about demo videos

We render a video/audio clip named *demo.mp4* attached to the complementary file. We render scene 3 with three different settings.

- Setting-1: emitter is directional and rotating
- Setting-2: emitter is at novel position and omnidirectional
- Setting-3: emitter is moving and directional

We demonstrate the effectiveness of Versa-SSL in setting-1 and Versa-ELE in setting-2 and setting-3.

## C   Social Impact

Our resounding framework promises to greatly enhance immersion and presence in AR/VR applications by delivering physically accurate, dynamic acoustics even when sound sources move freely through virtual spaces. However, the same technology could be misused to generate deceptive or manipulative audio-visual scenarios with "fake" acoustic environments that mislead users about their surroundings, or deep-fake conversations where background sounds betray false contexts. To mitigate this, a potential way is to inject an inaudible watermark into the audio to detect whether it is being used for malicious purposes.

