# OpenReview forum: "Resounding Acoustic Fields with Reciprocity"
_NeurIPS.cc/2025/Conference — NeurIPS 2025 poster_

### Official Review · Reviewer_sQL3 · 2025-06-21

**Clarity:** 2
**Significance:** 2
**Originality:** 3
**Rating:** 4
**Confidence:** 4

**Summary:**

This paper proposes training techniques to improve the performance of NNs for solving acoustic field learning problems under a particular situation where the source positions are sparse (which they call a resounding task).
- The basic idea is to train the NNs with an auxiliary loss term when learning an acoustic field so that its output closely follows to satisfy Rayleigh's reciprocity theorem.
- Two approaches are proposed: one that simply preserves the output when the source-receiver positions are switched (Versa-ELE), and one that decouples from the directivity pattern and trains to preserve the reciprocal relationship (Versa-SSL).
- The authors show that the Versa technique improves quantitative and qualitative evaluation performance on a variety of existing studies, including NAF, INRAS, AV-NeRF, and AVR.

**Questions:**

- Concerns about the relationship between Versa and the reciprocity

    - Versa is essentially a training technique that guides the NN output to remain similar when the source-receiver positions are switched. As the authors associate this technique with the ‘acoustic reciprocity principle’, this reviewer hopes that the paper will present results that show how this technique relates to reciprocity in practice. Just so there is no misleading the reader.

        1. The reviewer believes that the Rayleigh reciprocity theorem is built upon many assumptions, such as linear time-invariance (LTI), symmetric reflection, etc., and it seems that AcoustiX (and probably GWA too) might easily meet such assumptions, but would MeshRIR and RAF also do as well?
        2. Would there be any results on how the IRs are preserved in terms of the position swaps for various datasets?
        3. Could you provide any samples of how well the output IRs match when the source-receivers are swapped? (Or other results that can support the claim that NNs learn reciprocity through Versa?)

- Questions on the perceptual user study (L303 and Appendix B.2-3)
    1. How were the subjects recruited for the listening evaluation?
    2. Was IRB approval for this not required? (I see that the final checklist says N/A.)
    3. Did the subjects have sufficient experience and background in listening and evaluating spatial acoustics?
    4. I wonder if 15 subjects are a large enough number for a listening test to show a statistical significance.

**Minor comments.**
- The results in Table 6: the caption says it’s for RAF-Furnished, but the text says it’s RAF-EmptyRoom (L531).
- What do you mean by mentioning “Impulse response modeling has gained lots of attention due to its application in virtual experiences” in L81?

**Ethical Concerns:**

["NO or VERY MINOR ethics concerns only"]

**Final Justification:**

I appreciate the authors' detailed responses and their willingness to improve the manuscript with clearer statements.
Considering the authors' explanation that Versa aims to tackle the real-world scenario, my concern was then any demonstrations or references seem to be necessary that can support the feasibility of this approach, which assumes reciprocity as an underlying principle in the measured wave field. The authors expanded on this and showed their intention to avoid potential misunderstandings and overclaims. The amount of impact that this paper can bring to the NeurIPS community is considered moderate, as it is essentially a simple technique based on very well-known principles (both the reciprocity itself and the training technique). For these reasons, I am adjusting my score from 3 to 4.

**Limitations:**

yes

**Paper Formatting Concerns:**

No paper formatting issues

**Quality:**

3

**Strengths And Weaknesses:**

**Strengths.**
1. The main strength and contribution of this paper is that the authors redefine a specific challenge where the emitter locations are sparse and propose a simple technique that can alleviate the difficulty.
2. The authors applied their proposed technique to 4 prior studies and reported performance on 3+ datasets, showing consistent performance gains for different configurations.
3. It is worth noting that the datasets studied in the manuscript (including the appendix) cover diverse types of acoustic observations such as ray-based simulation, ray-and wave-based hybrid simulation, and real recording measurements.

**Weaknesses.**
1. In order for this study to be more convincing, it needs to be shown how the situation the authors envision — sparse emitter positions — is indeed a major challenge.
    - Given the IR obtained from computer simulations such as AcoustiX or SoundSpaces, it seems unlikely that emitter positions would have to be sparse, so I don't see much utility in the resounding task, and by extension Versa.
    - In a real-world setting where IR is measured, being able to predict the acoustic field with the least source positions is an advantage, but then, does reciprocity (or something to support Versa's validity to be associated with a reciprocity) hold in all real recordings covered in this study?
2. This reviewer is concerned that the method proposed in this paper is not sufficiently presented to be called reciprocity in practice.
    - The reciprocity principle is a fundamental concept, but it involves many assumptions.
    - It does not appear to be sufficiently demonstrated how well the datasets covered in this study meet those assumptions, and therefore how valid it is to apply this approach.

    (Please refer to the Question section below for some questions regarding this connection.)

---

> ### Author Rebuttal · Authors · 2025-07-30
>
> We thank Reviewer sQL3 for the careful review. We appreciate the reviewer for recognizing that our method shows “consistent performance gains for different configurations”, and our used datasets “cover diverse types of acoustic observations”. We address your concerns as follows.
>
> ---
> > **Review W1 part1: Question on Versa on Simulated dataset.**
>
> In the simulated environment, one can freely simulate the IRs at any position. The reason why we incorporate the simulated dataset in this paper is to demonstrate the effectiveness of our method in various dataset settings. We answer the rest of the questions (W1 part2, W2 and Q1) about whether reciprocity holds in the real world environment as follows.
>
> ---
> > **Review Q1(1), W2, W1 part2: Whether requirements for reciprocity are met in the real dataset?**
>
> The Rayleigh reciprocity theorem assumes linear, time-invariant (LTI) systems and symmetric propagation media. In a simulated environment, these could be easily satisfied through ray-based or wave-based methods. In real environment, most room acoustic environments used for impulse response measurements (e.g., MeshRIR, RAF) meet these criteria:
> 1. The environment is static during measurement and there is no moving object or human in the scene, making it time-invariant. (As also noted in the dataset collection procedure in the MeshRIR and RAF paper)
> 2. All common materials and structures (e.g., drywall, wood, concrete, furniture, carpeted floors) exhibit symmetric acoustic reflection behaviors. Only specially-engineered materials like metamaterials with time-varying impedance, active acoustic diodes, or isolators exhibit non-symmetric reflections. However, these materials are mostly found only in laboratories or engineered acoustic environments, and are not presented in daily life, especially in room acoustic measurement scenes.
>
> Based on the above analysis, real datasets and real-life scenarios would meet the requirements for reciprocity. We also show some RIR measurements in real-world datasets to prove reciprocity in the following answers.
>
> ---
> > **Review Q1(2), W2, W1 part2: Preservation of IRs after position swaps in real dataset.**
>
> It is easy to verify the reciprocity in the simulated dataset. For this question, we focus on proving reciprocity on the real dataset.
>
> **Discussion on MeshRIR and RAF:**
>
> MeshRIR has different areas for emitter/listener positions, and thus does not include RIR pairs that have the same emitter/listener positions after swapping. RAF has significantly different emitter and receiver gain patterns, nor do they have listener orientation measurements. Thus, there isn’t any RIR pair that have the same emitter/listener positions and orientations after swapping. As a result, we could not empirically evaluate the impulse response invariance for these two datasets based on their current data points. However, experimental results in our original paper show consistent improvements of Versa on them, indicating reciprocity constraints do hold for them.
>
> **Discussion on our collected real dataset as a proof:**
>
> Though we can not get the pairs of IRs after switching the position of the listener/emitter in the MeshRIR and RAF dataset (these samples are not included in the dataset), we collect our own real dataset as proof.
>
> **Dataset collection procedure:** We conducted an experiment in 3 various indoor environments (an office room(9m x 8m, a conference room (13m x 7m), a kitchen (5m x 9m)) using a microphone (ADMP401 MEMS microphone) and a speaker (Bose SoundLink Revolve II speaker) that are both omnidirectional. They are wired connected to an audio interface (BELA board), which can synchronize between the playing and recording. We follow the data collection in DiffRIR and play a 10-second chirp signal that spans 50Hz to 20KHz to collect RIR samples. We collected the impulse responses emitted from position A to B and from position B to A. Under the reciprocity principle, these pairs of impulse responses should be identical. Positions A and B are randomly chosen to represent the diverse acoustic fingerprint in an environment. We collect 20 such RIR pairs in each environment, placing them in a diverse combination of speaker and microphone positions.
>
> **Results:** We measure the similarities within pairs of RIRs that have swapped emitter/listener positions and the cross-paired RIRs. As shown in the table, in each of the scenes, the paired RIRs have much lower acoustic metrics (some metrics are x10 smaller) compared with cross-paired RIRs. This confirms that the RIRs are preserved in terms of the position swaps for various real-world environments.
>
> |                               |  Amp |  Env |  T60 |  C50 |  EDT  |
> |-------------------------------|:----:|:----:|:----:|:----:|:-----:|
> |     Kitchen cross-paired RIRs     | 1.74 | 12.0 | 11.5 | 3.69 |  89.1 |
> |          Kitchen paired RIRs         | 0.24 |  1.8 |  2.1 | 0.29 |  9.8  |
> | Conference room cross-paired RIRs | 1.09 |  7.5 | 18.7 | 3.35 |  120  |
> |  Conference room paired RIRs  | 0.22 |  1.0 |  3.3 | 0.23 |  11.9 |
> |   Office room cross-paired RIRs   | 1.54 | 16.5 | 24.5 | 2.49 | 104.5 |
> |    Office room paired RIRs    | 0.23 |  2.0 |  2.2 | 0.18 |  11.0 |
>
> We will also incorporate these supplementary results in the revised version to show the reciprocity for the real-world environment.
>
> ---
> > **Review Q1(3), W2, W1 part2: Provide evidence of NN have learn reciprocity with Versa.**
>
> We acknowledge the importance of validating whether the neural acoustic field models trained with Versa indeed exhibit reciprocity. We included a similar experiment in Q1(2). We randomly query the trained AVR and NAF models on swapped emitter/listener pairs at each scene and evaluate the predicted impulse response similarity. As shown in the table, the predicted RIRs pairs with Versa training exhibit much lower acoustic metrics (showing higher similarities for paired predictions after position swap) compared with the neural acoustic field without the Versa training. These show that the NN models learn reciprocity through the Versa strategy. In addition, we promise to include more waveform visualizations of the model-predicted RIRs in the final manuscript.
>
> |                               |  Amp |  Env |  T60 |  C50 |  EDT |
> |-------------------------------|:----:|:----:|:----:|:----:|:----:|
> | AVR w/o Versa-ELE | 0.54 | 10.6 | 16.6 | 4.13 | 86.9 |
> |  AVR w/ Versa-ELE  | 0.06 |  0.5 |  4.8 | 0.61 | 10.8 |
> |       NAF w/o Versa-ELE       | 0.33 |  5.9 | 13.5 | 2.83 | 50.4 |
> |        NAF w/ Versa-ELE       | 0.27 |  3.0 |  5.4 | 0.45 |  8.4 |
>
>
> ---
> > **Review Q2: Questions on the perceptual user study.**
>
> - **IRB and Recruitment:** We have consulted with our host institution's IRB office regarding the ethical requirements of our audio listening evaluation. Based on the nature of the study that involves only anonymous adult participants, no collection of personal or identifiable data, and no procedures involving risk or discomfort,  the study was formally deemed exempt from IRB review and granted a waiver with “non-human subject determination”. We will clarify this waiver status in the revised paper to avoid confusion.
>
> - **Background:**  Though the participants were not expert acousticians, we ensured they had sufficient familiarity with spatial audio experiences. We included a brief session before the evaluation to inform them about the process of the audio evaluations.
>
> - **Sample size:** Recent related works like [R1] use 12 users for their perceptual study. We understand your concern and further extend our user study from 15 users to 26 users with an updated result shown below. Our method still demonstrates superior perceptual preference across all aspects.
>
> |  | Versa-ELE vs Vanilla | Versa-SSL vs ELE |
> |:-----------:|:--------------------:|:----------------:|
> |    Volume   |  88.5% | 86.5%  |
> | Directional |  88.5% | 88.5%  |
> |   Overall   | 92.3%  | 80.8% |
>
>
> - **Additional evaluations:** We also incorporate an objective, reference-based audio similarity metric Deep Perceptual Audio Metric (DPAM) as suggested by reviewer eBxj. This metric can evaluate the audio similarity close to human perception. We evaluate the similarities between the rendered sound (IR convolve with music) to the ground truth. As shown in the table, Versa-ELE reduces DPAM error by 20%, and Versa-SSL further reduces error by 36% compared with Versa-ELE on the AVR method. Versa-ELE also boosts the performance of other methods by 18%.  These show our method has stronger perceptual similarity to ground truth. These results are consistent with the user study findings and reinforce the perceptual benefit of our method in the resounding task.
>
> | Method | DPAM score |
> |:-------:|:------------:|
> |   NAF + Versa-ELE   | 1.53 -> 1.26 |
> |  INRAS + Versa-ELE  | 1.42 -> 1.19 |
> | AV-NeRF + Versa-ELE | 1.47 -> 1.15 |
> |   AVR + Versa-ELE   | 1.44 -> 1.17 |
> |   AVR + Versa-SSL   | 1.44 -> 0.86 |
>
> We will revise the manuscript to clarify the study design, IRB waiver and add the new evaluation metric to enhance the robustness of our perceptual analysis.
>
> [R1] Tong, G., Leung, J. C. H., Peng, X., Shi, H., Zheng, L., Wang, S., ... & Chakravarthula, P. (2025). Multimodal Neural Acoustic Fields for Immersive Mixed Reality. IEEE Transactions on Visualization and Computer Graphics.
>
> ---
> > **Minor comments:**
>
> **Typos:** Thanks for pointing out the typos and we will fix them in the revised version.
>
> **What does L81 mean?** It highlights the growing importance of RIR modeling in immersive audio rendering for virtual and augmented reality applications. RIR characterizes how sound propagates through an environment, which is essential for generating spatially and temporally realistic audio cues. These cues greatly enhance user immersion in virtual environments, aligning with the motivation behind our work on neural acoustic fields and the resounding task.

---

> > ### Comment · Reviewer_sQL3 · 2025-08-01
> >
> > Thanks for the response. While I appreciate the authors for sharing the additional experiment results, some of my concerns have not been fully resolved:
> >
> > ****
> >
> > **Review Q1(1), W2, W1 part2: Whether requirements for reciprocity are met in the real dataset?**
> >
> > I understand the underlying assumptions for the reciprocity theorem. However, I remain cautious about making strong claims that applying reciprocity to the methodology proposed in this paper is *natural* or that the dataset used *satisfies* reciprocity.
> >
> > - This is because neither the manuscript nor the authors' rebuttal provides any prior research supporting these claims.
> >
> > - To respond line-by-line on the authors' response:
> >     > In a simulated environment, these could be easily satisfied through ray-based or wave-based methods. In real environment, most room acoustic environments used for impulse response measurements (e.g., MeshRIR, RAF) meet these criteria:
> >
> >     - Even simulations based on linearized wave equations and simple impedance-based reflection modeling are subject to some degree of numerical error, which can lead to a breakdown in these assumptions. And it's widely understood that measured data can be sensitive to such noise, even when measured in very simple environments, as microphones with perfectly linear responses are rare.
> >     - It is, obviously, *not* the reviewer's intention to ask the authors to demonstrate that reciprocity is satisfied "despite" the difficulties. However, it is advised to point out in their manuscript that these difficulties exist and demonstrate how well their simulators and data meet them, as it is important to avoid the misconception that reciprocity applies to *any* acoustic models when they have omnidirectional emitter/receiver.
> >
> >     > All common materials and structures (e.g., drywall, wood, concrete, furniture, carpeted floors) exhibit symmetric acoustic reflection behaviors. Only specially-engineered materials like metamaterials with time-varying impedance, active acoustic diodes, or isolators exhibit non-symmetric reflections. However, these materials are mostly found only in laboratories or engineered acoustic environments, and are not presented in daily life, especially in room acoustic measurement scenes. Based on the above analysis, real datasets and real-life scenarios would meet the requirements for reciprocity.
> >
> >     - It is a minor concern, but to mention on the claim above: many materials encountered in everyday life, such as rubber, thin films, and porous media like sponge or carpet, *possess* nonlinear response characteristics. However, because the sound pressure fluctuations people encounter in daily life are not significant ($\leq$ 100 dB), most of these characteristics do not differ significantly from (quasi-)linear response characteristics. [1]
> >
> > - Overall, until the paper rigorously demonstrates to what extent reciprocity is actually satisfied, or provides references that support that conclusion, I find it difficult to make a strong claim that this paper fully "enforces reciprocity."
> >
> >
> > **Minor comment: What does L81 mean?**
> > > It highlights the growing importance of RIR modeling in immersive audio rendering for virtual and augmented reality applications. (...)
> >
> > Modeling RIR for immersive spatial audio has been around for over 50 years [2], and such an approach can be considered quite old, considering that the practical FFT was implemented around 1965 (Cooley and Tukey). In particular, among the various RIR modeling methods, the *convolutional approach* (which is the type discussed in this paper) is one of the simplest and is becoming not quite commonly used in modeling high-quality RIRs. I wonder if the authors' expressions of "gained lots of attention" or "growing importance" for RIR have a particularly significant meaning.
> >
> > ****
> >
> > The reviewer appreciates the authors' presentation of additional experimental results and hopes to include the relevant table in the revised version. Yet, some concerns regarding potential misleadings in this manuscript remain unresolved. More specifically, the statement that 'the proposed Versa-ELE leverages reciprocity' delivers the impression that the reciprocity is assumed in the data, which seems to require further refinement or supporting evidence. The reviewer is not fully convinced that the manuscript's rigor is sufficiently refined to meet the publication quality.
> >
> > ****
> >
> > [1] Hamilton, M. F., & Blackstock, D. T. (2024). Nonlinear acoustics (p. 447). Springer Nature.
> >
> > [2] Valimaki, V., Parker, J. D., Savioja, L., Smith, J. O., & Abel, J. S. (2012). Fifty years of artificial reverberation. IEEE Transactions on Audio, Speech, and Language Processing, 20(5), 1421-1448.

---

> ### Author Response · Authors · 2025-08-03
> **Further reply part 1**
>
> (This is the first part of the response)
>
> Dear reviewer, thank you for your thoughtful and detailed follow-up comments and especially your appreciation about our additional experimental results. We are very grateful for your feedback in helping us improve the scientific clarity and rigor of our work. Below, we further address your concerns and propose concrete revisions to the manuscript.
>
>
> **Clarification on reciprocity assumptions and dataset validity**
>
> As the reviewer suggests, we do agree that the perfect reciprocity of RIRs recordings can not be satisfied in the real-world. Thus the strict reciprocity of RIRs cannot be assumed or proved in our selected dataset. Instead of proving or assuming perfect reciprocity in the real-world, our method seeks to leverage reciprocity-inspired inductive bias in modern neural acoustic field training. We aim to solve potential misunderstandings and avoid overclaim, providing the detailed reasoning as follows.
>
> Reciprocity of the acoustic system was first proposed by Helmholtz [R1] and expanded by Rayleigh in the theory of sound [R2]. As a theoretical principle, it relies on assumptions such as linearity, time-invariance, and symmetric propagation. As you pointed out, we understand that in real-world recordings, perfect global reciprocity, i.e. impulse response invariance after switching emitter and listener positions, is hard to achieve due to all factors including device non-linearity, non-linear material responses, hardware thermal noise. We will also cite relevant work [R3] to acknowledge the non-ideal behaviors of some common materials under high sound energy, while noting that in the low-intensity, indoor acoustic settings, quasi-linear response is a widely accepted approximation. We also note that, as you suggested, even in a simulated environment, the reciprocity does not strictly hold due to the numerical calculation errors, randomness in the ray-tracing and so on.
>
> In fact, as also presented in our additional real-world RIR collection experiments, the distance between the paired RIRs is not strictly zero, the recording is naturally influenced by the hardware thermal noise and all the non-perfect properties in the real world mentioned before. However, the paired RIRs still have much higher similarities compared with the cross-paired RIRs. To validate the reciprocity on the simulated dataset, we also include an additional experiment on RIRs similarities on the AcoustiX dataset. Results show that with the increased number of rays, the error introduced by the numerical results on the reciprocity gets decreased since there is less randomness in the higher order reflections in the ray tracing procedure.  Overall the results indicate that the RIRs are very close(100X) after emitter/listener swapping with 10^6 rays on the simulated dataset. Again the RIRs are not fully identical (metrics are not 0), but through the waveform visualization (which we will incorporate in the revised version) we found that the RIRs are nearly the same. We also adopt this 10^6 rays setting in the simulation settings. Further increasing this number in the simulation would cause very high computational and memory overload.
>
> |                       |  Amp |  Env |  T60 |  C50  |  EDT |
> |-----------------------|:----:|:----:|:----:|:-----:|:----:|
> |     unpaired RIRs     | 0.43 |  5.2 | 16.6 | 1.953 | 98.5 |
> | Paired RIRs 10^4 rays | 0.13 | 0.14 |  5.9 |  0.4  |  7.1 |
> | Paired RIRs 10^5 rays | 0.07 | 0.07 |  2.1 |  0.12 |  3.8 |
> | Paired RIRs 10^6 rays | 0.06 | 0.03 | 0.48 | 0.045 | 0.82 |
>
> We also understand your concern about whether perfect reciprocity holds on the impulse response. While the linearity, symmetric propagation and other requirements are needed to satisfy the global reciprocity and the impulse response invariant features, real world collection can inevitably encounter factors like noise and non-linearity, resulting in imperfect impulse response that could “break” the reciprocity. Prior work has demonstrated that reciprocity may not hold perfectly due to speaker imperfections. However, with proper measurement techniques and deviation correction, reciprocity remains valid and reliable in real-world acoustic environments, even in reverberant settings [R6]. Suggested by [R4], even when slight disturbances exist, reciprocal measurement setups still yield data with acceptably small uncertainty. Also the acoustic path reciprocity has facilitated the impulse response measurements by allowing for switching the positions of emitter and listener without affecting the measurement, thereby enabling measurement scenarios where loudspeaker placement is potentially challenging and microphone placement is straight forward [R5]. A similar idea has also been leveraged in our paper to introduce the reciprocity inspired Versa method as a learning strategy.

---

> ### Author Response · Authors · 2025-08-03
> **Further reply part 2**
>
> (Continue of the previous discussion, this is the part two of the response)
>
> We do agree that the perfect reciprocity of RIRs can not be satisfied in the real-world. And we aim not to make the strong assumption or claim that reciprocity is naturally applied to any real dataset. To avoid this overclaiming, we will revise our wording throughout the paper to clarify that Versa does not assume strict reciprocity in the data, but rather leverages reciprocity-inspired constraints as an inductive bias in modern neural acoustic field training. Our empirical results also show that with our reciprocity inspired Versa method, each model can get better performance in the acoustic field reconstruction. This shows that injecting these physical properties can make the NN model learn a more generalizable acoustic field. In addition, we will replace phrases like "enforces reciprocity" with "encourages reciprocity-based consistency" or "incorporates reciprocity as a soft constraint.” Finally, as also noted in [R4], the reciprocity of a practical real system needs to be checked to eliminate the doubt. We would also add text to advise that before using Versa on the dataset, one can first check if the reciprocity of RIRs exist in their setting first.
>
> Furthermore, we will add the following clarifying paragraph in our method section and introduction:
>
> _"While the principle of reciprocity holds under idealized conditions (e.g., linear, time-invariant media with symmetric reflection), real-world environments and simulated systems may deviate from these assumptions due to many factors like measurement noise, nonideal hardware responses, or complex material properties and so on. Rather than assuming perfect reciprocity in the data, Versa uses reciprocity as a guiding principle to regularize the learning process by encouraging output consistency when emitter and listener positions are exchanged. This approach allows the model to benefit from symmetry in wave propagation, even when perfect reciprocity does not strictly hold."_
>
>
> **Further clarification of our motivations at Line 81**
>
> We appreciate the reviewer’s historical perspective. We agree that convolution with an impulse response has been a foundational rendering technique for over 50 years, and it remains the physically correct way to simulate how sound propagates through a linear, time-invariant medium. Our intention with this statement was to highlight the renewed interest in the way to render impulse response within the deep learning and neural acoustic rendering topics (NAF and INRAS (NeurIPS’22); AV-NeRF (NeurIPS’23), DiffRIR (CVPR’24), AVR and AV-Cloud (NeurIPS’24), NeRAF (ICLR’25), and more papers on recent ICCV, ICML submissions), especially in the context of spatial audio synthesis for immersive experiences. We will rephrase the sentence to better reflect this nuance:
>
> _"Impulse response modeling has seen renewed interest in recent years, particularly in the context of neural acoustic field learning and immersive spatial audio rendering for virtual environments."_
>
>
> **Summary with proposed manuscript revisions:**
>
> Again we would like to thank you for your engagement to help us improve the scientific rigor of our paper. To address the your key concern, we will make the following changes in the revised manuscript:
> - Adjust phrasing throughout to reflect that Versa uses reciprocity as a soft inductive bias.
> - Add a subsection discussing the assumptions and limitations of reciprocity in real/simulated datasets.
> - Include references to acoustic modeling literature acknowledging material and device nonidealities. We also will add more references to discuss the reciprocity in a practical real-world acoustic system.
> - Add the additional experimental results to demonstrate the RIRs similarities of paired one and cross-paired one both in our self-collected real dataset and the simulated dataset.
> - Clarify our terminology around RIR modeling’s historical context and our paper’s motivations.
>
>
> **References**
>
> [R1] Helmholtz, H. von. "Theorie der Luftschwingungen in Röhren mit offenen Enden." (“Theory of air vibrations in € pipes with open ends”) (1860): 1-72.
>
> [R2, cited in 38 in the manuscript] Rayleigh, John William Strutt Baron. The theory of sound. Vol. 2. Macmillan, 1896.
>
> [R3] Hamilton, M. F., & Blackstock, D. T. (2024). Nonlinear acoustics (p. 447). Springer Nature.
>
> [R4] Ten Wolde, Tjeert. "Reciprocity measurements in acoustical and mechano-acoustical systems. Review of theory and applications." Acta Acustica united with Acustica 96, no. 1 (2010): 1-13.
>
> [R5] Baba, Youssef EI. Room Geometry and Low-Frequency Transfer Function Inference Using Acoustic Transducers. Friedrich-Alexander-Universitaet Erlangen-Nuernberg (Germany), 2022.
>
> [R6] Lu, Ling, Hongling Sun, Ming Wu, Qiaoxi Zhu, and Jun Yang. "A modified electro-acoustical reciprocity method for measuring low-frequency sound source in arbitrary surroundings." Applied Acoustics 116 (2017): 1-8.

---

> > ### Comment · Reviewer_sQL3 · 2025-08-05
> >
> > I appreciate the authors' detailed responses and their willingness to clarify the manuscript. My concerns have been largely addressed, and I am adjusting my score from 3 to 4.

---

> > > ### Author Response · Authors · 2025-08-05
> > >
> > > We would like to thank you for your active engagement in ths discuss phase. Your suggestions have helped us to improve the rigor and clarity of the manusciprt a lot. We will revised the manuscript as suggested. Thank you!

---

### Official Review · Reviewer_tDBR · 2025-07-02

**Clarity:** 3
**Significance:** 2
**Originality:** 3
**Rating:** 4
**Confidence:** 2

**Summary:**

The paper presents Versa, a physics-inspired approach to learning acoustic fields, and shows how this approach can be used to improve on existing methods for "resounding" tasks analogous to "relighting" in computer graphics.

**Questions:**

I watched and listened to the demo in the supplementary materials (the video was labeled "CVPR_25" – I suggest relabeling it ;)) which was helpful, but as a non-expert, I didn't know what to listen for to judge the quality and really understand the comparison with baselines.  Having some helpful pointers (listen for this, compare this) woud help the NeurIPS community better appreciate the value of this paper.

All the demos involved music, and all music in a pretty limited span of pop genres.  How does the approach generalize to other kinds of music, and other kinds of sounds.  What about speech?

How does the approach interact with physical models one has (built or learned) for material that surfaces in the environment are made out of (e.g., wooden floor, stone floor, plaster walls, carpet, etc.).  I expect these would change the acoustics of an environment substantially?  Is there a way the approach extends to include models of or knowledge about the acoustically relevant physics of the environment?

Given the paper's limited accessibility outside of its specialty area, and limited appeal to the broader NeurIPS community, could the authors draw out a broader message or lesson of interest that readers could take home, beyond just that this seems to work pretty well?

**Ethical Concerns:**

["NO or VERY MINOR ethics concerns only"]

**Final Justification:**

The rebuttal helpfully addressed my concerns and I think their proposed paper changes will make the paper stronger.  I still feel unsure whether NeurIPS is the right venue for this work so I'll maintain my score of borderline accept.  This is mostly about my own lack of familiarity with this area of AI.  The "accept" criterion say: "Technically solid paper, with high impact on at least one sub-area of AI or moderate-to-high impact on more than one area of AI".  I'm just enough of an expert to judge that this work meets those criteria.

**Limitations:**

yes

**Quality:**

3

**Strengths And Weaknesses:**

I have to start off by stating that this paper is far from my expertise.  At a high level, the problem seems interesting, and relevant to a range of computer graphics and interactive (VR, game) design settings.  The combination of physical modeling sophistication with machine learning sophistication is a strength of the paper.  The improvements over baselines appear substantial, but it's hard for me to judge the quality or significance as a nonexpert.

I don't see technical weaknesses but I have low confidence in my ability to detect these given that I'm not an expert in this paper's topic.  As a NeurIPS submission, I do view it as a weakness of the paper that it has limited accessibility and appeal to the broader NeurIPS community outside of those interested in machine learning for audio representations.   I'm not sure why the paper isn't better suited for SIGGRAPH or a computational audio or acoustics venue.

---

> ### Author Rebuttal · Authors · 2025-07-25
>
> We thank Reviewer tDBR for the constructive suggestions. The feedback has significantly contributed to the improvement of our paper to make it better accessible to the broader community. We appreciate the reviewer for recognizing that “the combination of physical modeling sophistication with machine learning sophistication is a strength”, and “the improvements over baselines appear substantial”, giving us a positive rating. We address your concerns as follows.
>
>
> ---
> > **Reivew W1: Concern about the scope and accessibility.**
>
> We understand the reviewer’s concern about accessibility to the broader NeurIPS community. While the application domain is audio and acoustics, our contribution is fundamentally about machine learning under a structured physical prior. Specifically, we propose reciprocity-grounded Versa as a learning strategy. We show the effectiveness of Versa in many existing work on audio field reconstruction that are published in the NeurIPS community (NAF, INRAS, AVR, and AV-NeRF). We also provide theoretical grounding and empirical evidence for the proposed method for learning the dynamic audio field.
>
> We hope the broader NeurIPS audience finds value in how our method bridges wave propagation and ML training pipelines. We will highlight these broader themes and revise the introduction and discussion to explicitly call out these cross-domain insights.
>
> ---
> > **Reivew Q1: Adding pointer in the demo.**
>
> Thanks for the advice, we will add extra pointers to highlight the comparisons and point-by-point comparison for the audio after releasing our work on the project webpage.
>
> ---
> > **Reivew Q2: Question about the rendered sound, how about the rendering speech?**
>
> We appreciate this thoughtful question. Our method fundamentally models impulse responses, which characterizes how sound waves propagate in the environment and heard by human ears. Since the impulse responses are independent of the input dry sound, it can be convolved with any sound source like music, speech and ambient noise. As a result, the resounding capability generalizes naturally across genres and source types.
>
> While we used music in the demos for its perceptual richness and clarity in spatial cues, our method works equally well with speech and other sounds. We will include additional speech-based examples in the supplementary material and project website to illustrate this.
>
> ---
> > **Reivew Q3: How does the approach interact with the physical model?**
>
> The physical characteristics of surfaces like sound reflections directly influence the room’s impulse response. Integrating explicit material priors, either from semantic scene understanding or estimated material parameters is a promising direction for future work. This could further enable data efficiency by grounding the acoustic model in known physical properties. Given the material’s properties, one can use a ray tracing method to render the impulse response, where the reciprocity property is already considered.
>
> Versa is complementary to these efforts. While material-based models focus on surface-level interactions, our method emphasizes the broader wave propagation paths, enforcing physical consistency across emitter-listener pairs. It leverages reciprocity to capture the cumulative effects of symmetric path interactions along surfaces for better acoustic field reconstruction. Besides, some of our baseline models (AV-NeRF, INRAS, NeRAF) include relevant physics of the environment from vision or 3D inputs. Experiment results show that Versa improves their performance substantially as well.
>
> ---
> > **Reivew Q4: Broader take home message.**
>
> We thank the reviewer for encouraging us to better articulate the broader message. At a high level, our work demonstrates how physical priors like reciprocity can be converted into effective training strategy, especially when training samples are sparse. Within the audio domain, we show the effectiveness in many works published in the NeurIPS community.
>
> Beyond the audio domain, this principle extends beyond acoustic signal to vision (light transport), radio frequency (RF) signal and so on. Similar ideas can guide neural rendering models to enforce multi-view consistency. For RF signals, electromagnetic reciprocity holds in many indoor environments. This can also be exploited to simulate virtual transmitters/receivers and reduce measurement overhead in RF channel modeling.
>
> Our key takeaway is the potential of incorporating physical principles into the training of neural networks. This approach not only reduces the 'black box' nature of the models but also enhances their generalization capabilities, often requiring less training data.

---

> > ### Comment · Reviewer_tDBR · 2025-08-05
> >
> > Thanks for these helpful responses.  I am glad the authors are planning to incorporate these thoughts into the revision of their paper, and it should be stronger as a result.

---

> > > ### Author Response · Authors · 2025-08-08
> > >
> > > Dear reviewer, we would like to thank you for your suggestion to broader our influence out of the acosutic community to even broader community in the neurips. Thanks for your positive rating.

---

### Official Review · Reviewer_eBxj · 2025-07-04

**Clarity:** 3
**Significance:** 3
**Originality:** 3
**Rating:** 4
**Confidence:** 4

**Summary:**

The proposed work aims to estimate the acoustic field of scenes that are dynamic such that there are sparse multiple emitters in the scene and these could be relocated. To address this, the authors propose a method based on the principle of wave reciprocity Emitter Listener Exchange (Versa-ELE) which allows for swapping the roles of emitters and listeners if the path between them remains unchanged, i.e. the gain pattern between them is unchanged when the direction is being swapped. To further extend the method to cases where emitters and receivers have different directional characteristics, the authors propose Versa-SSL, a self-supervised learning strategy that encourages consistent predictions between swapped roles in conjunction with Acoustic Volume Rendering (AVR). Evaluations on simulated dataset of AcoustiX-Diff and real datasets (Mesh-RIR and RAF) indicate that incorporating ELE augmentation or SSL training with AVR  improves performance measured with multiple acoustic metrics.

**Questions:**

I would appreciate authors’ response to the following questions:

Q1. I was not able to find the expression of the loss used for SSL (i.e. L_{a-ssl}). In general, the SSL paradigm is not fully clear. For example I am not sure if negative pairs are used as contrastive loss and how these are generated.

Q2. Deep Perceptual Audio Metric (DPAM) is a useful metric calibrated with human audio perception (R6). How would Versa-ELE and -SSL perform for this metric?

Q3. Related to W1-a, how would Versa-ELE and -SSL compare with BEE and can ELE approach be implemented in conjunction with BEE?

R6. Manocha, P., Finkelstein, A., Zhang, R., Bryan, N. J., Mysore, G. J., & Jin, Z. (2020). A differentiable perceptual audio metric learned from just noticeable differences. arXiv preprint arXiv:2001.04460.

**Ethical Concerns:**

["NO or VERY MINOR ethics concerns only"]

**Final Justification:**

The work proposes a new reciprocity-inspired learning approach for acoustic field estimation of a scene. In particular, the proposed loss (reciprocity contrastive loss) can be combined with various acoustic field estimation methods and enhance their accuracy. Experiments that the authors conducted in the manuscript and additional results on effectiveness of the reciprocal contrastive loss that the authors stated in the rebuttal and discussion of generalization, indicate an effective and useful method for improving acoustic field estimation and serve as a foundation for further extensions. I am supportive of publication of the work, given that the authors incorporate the revisions presented in the rebuttal in the revised version of the paper, and raising my score for this paper to 4.

**Limitations:**

yes

**Quality:**

3

**Strengths And Weaknesses:**

Strengths:

S1. The paper addresses an important problem of dynamic scenes acoustic field reconstruction, which many approaches in acoustic field research are unable to fully address at the onset.

S2. The proposed ELE approach makes use of wave propagation reversibility and proposes an augmentation approach to switch the roles of emitters and listeners, requiring that the paths from emitters to listeners and backward will be preserved. Such augmentation can be combined with existing approaches for acoustic field reconstruction.

S3. The work proposes a self learning (SSL) approach for further precision / realistic cases. where the acoustic field prediction is constrained to be identical under switching the role of the emitter and listener creating positive pairs for SSL training.

S4. Results with several acoustic field reconstructions baselines show improvement in multiple metrics.

Weaknesses:

W1. The related work is not up to date and in general I found the related work section to be too scarce. In particular, two major concerns

The authors omitted work that addressed dynamic acoustic field reconstruction. e.g. R1 where acoustic field reconstruction for such scenes was considered with the use of sparse stationary receivers placed in the scenes, making the paths from dynamic emitters to listeners to be emitters -> receivers-> listeners.

Several recent acoustic field approaches were not included in related work and results, e.g. INRAS++ (R2), AVCloud (R3), NeRAF (R4), etc.

R1. Chen, M., Su, K., & Shlizerman, E. (2023). Be everywhere-hear everything (bee): Audio scene reconstruction by sparse audio-visual samples. In Proceedings of the IEEE/CVF International Conference on Computer Vision (pp. 7853-7862).

R2. Chen, Z., Gebru, I. D., Richardt, C., Kumar, A., Laney, W., Owens, A., & Richard, A. (2024). Real acoustic fields: An audio-visual room acoustics dataset and benchmark. In Proceedings of the IEEE/CVF Conference on Computer Vision and Pattern Recognition (pp. 21886-21896).

R3. Chen, M., & Shlizerman, E. (2024). AV-Cloud: Spatial Audio Rendering Through Audio-Visual Cloud Splatting. Advances in Neural Information Processing Systems, 37, 141021-141044.

R4. Brunetto, A., Hornauer, S., & Moutarde, F. (2024). NeRAF: 3D Scene Infused Neural Radiance and Acoustic Fields. arXiv preprint arXiv:2405.18213.

W2. The results were not applied to same datasets as the baselines, i.e. SoundSpaces, on Replica, Matterport3D, and on a moderate number of scenes.

---

> ### Author Rebuttal · Authors · 2025-07-30
>
> We thank Reviewer eBxj for the thoughtful review. The constructive feedback has significantly contributed to the improvement of our paper. We appreciate the reviewer for recognizing that our work deals with a problem that prior methods “are unable to fully address at the onset”, “can be combined with existing approaches”, and “show improvement in multiple metrics”. We address your questions and concerns one by one as follows.
>
> ---
> > **Review W1 part1. Discussion on the related paper BEE.**
>
> We thank the reviewer for pointing out this relevant line of work on dynamic acoustic field reconstruction. We agree that BEE[R1] is an important contribution in the broader space of learned acoustic field modeling. While we did not include this work in our original related work section, we will add it in the revised manuscript and clarify how our setting differs as follows.
>
> 1. **Training Setup**: BEE trains on a large-scale, multi-scene corpus with dense emitter-listener pair sampling across a variety of environments. It also leverages audio-visual correspondence as a strong learning signal. After training on the large-scale dataset, it can leverage the sparse listener samples in the novel scene to reconstruct the acoustic field. In contrast, our method focuses on per-scene optimization, only trained on a sparse set of emitter positions, and does not consider cross-scene generalization. Our goal is to reconstruct a physically consistent acoustic field within a single environment, given limited emitter setups.
>
> 2. **Task definition and output**: BEE targets novel scene generalization, which involves predicting audio waveforms or spectrograms for novel listener positions in the environment given sparse receiver samples. In contrast, our work tackles the resounding task in the same acoustic scene, which focuses on estimating the impulse response at an arbitrary emitter/listener location within a scene. This allows for more flexible downstream rendering.
>
> We will revise the related work section to include and discuss our differentiation from BEE and similar approaches.
>
> ---
> > **Review W1 part2: Discussion on the related paper AVCloud, NeRAF, INRAS++**
>
> Thank you also for pointing out recent advances in acoustic field modeling. We now include two other open-source papers, including AV-Cloud (R3) and NeRAF (R4), as additional baselines in our experiments. Both the AV-Cloud and NeRAF leverage 3D visual information for acoustic prediction.
>
> We experimented with the AcoustiX-Same dataset and applied our Versa-ELE method on top of AV-Cloud and NeRAF. We find that Versa-ELE improves both of their performance by a large amount, further supporting the generality of our approach to various acoustic field approaches.
>
> The INRAS in our baseline model is already implemented as INRAS++ (R2), extending the 3d grid sampling and extra loss components. The other baseline NAF and AVNeRF implementations also follow the RAF paper (R2).
>
> We will cite all of these works (R1–R4) and clarify the baseline settings in the revised version, and expand our related work section to better reflect recent progress in the field.
>
> |                   | STFT |  Amp | Env |  T60  |  C50 |  EDT |
> |-------------------|:----:|:----:|:---:|:-----:|:----:|:----:|
> |       NeRAF       | 2.08 | 0.50 | 4.6 |   20.5  |  4.54 | 33.2 |
> | NeRAF + Versa-ELE | 1.82 | 0.35 | 4.0 |  16.9 | 3.08 | 23.4 |
> |      AVCloud      | 1.87 | 0.48 | 3.7 |  16.6 | 3.66 | 32.3 |
> | AVCloud+Versa-ELE | 1.26 | 0.32 | 2.7 |  13.2 | 2.07 | 15.8 |
> |        AVR        | 2.34 | 0.51 | 3.2 | 21.0 | 2.82 | 31.9 |
> |     AVR + ELE     | 1.20 | 0.25 | 2.1 |  13.5 | 1.75 | 15.1 |
>
> ---
> > **Review W2 part1: Evaluation dataset, “Not applied to the same dataset as the baselines, like SoundSpaces”.**
>
> Thank you for raising the dataset issue. For the SoundSpaces dataset, we also noted some time-of-flight errors on the simulated impulse response as prior paper (AVR), which makes it not ideal to test on some models like DiffRIR and AVR. As a result, we use other simulation engines (AcoustiX and GWA) and datasets, plus real datasets (MeshRIR and RAF), with diverse settings to evaluate the model’s performance.
>
> ---
> > **Review W2 part2: Concern on the number of evaluation scenes.**
>
> As pointed out by the Reviewer sQL3 in the strengths section, we used various simulated datasets, including ray-based, wave and ray hybrid simulation datasets, and also real datasets to test our method. Our evaluation includes various simulated scenes (3 for AcoustiX, 3 for GWA) and real scenes (MeshRIR, 2 RAF scenes), with a total of 9 scenes. Our experiment also involves various setups for gain patterns, resulting in more than 15 different scene+gain pattern combinations for evaluations. We also incorporate a diverse number of emitters/listeners in each scene to evaluate the robustness of our method. Through these diverse evaluations, we have verified the effectiveness of our Versa framework in various models that purely rely on neural acoustic feature (NAF), scene geometries (INRAS), acoustic propagation phenomenon (AVR), visual-acoustic correspondence (AV-NeRF, AVCloud, NeRAF).
>
> ---
> > **Review Q1: Confusion on the self-supervision loss.**
>
> We do not involve negative pairs for self-supervised learning as common contrastive loss. For our self-supervision loss, let $h_1 = F(P_l, P_e, w_l, w_e, G, G)$ and $h_2 = F(P_e, P_l, w_e, w_l, G, G)$. The SSL loss $L_{a-ssl}$ only forces $h_1$ and $h_2$ to be identical. For the simplest version, we can directly optimize on the L1 loss between the waveform $h_1$ and $h_2$, i.e., $L_{a-ssl} = |h_1 - h_2|$. In our implementation, we follow a similar strategy to calculate the L1 distance between $h_1$ and $h_2$ in time and spectrogram domain as in the previous framework.
>
> ---
> > **Review Q2: Include DPAM metric for evaluations.**
>
> Thank you for your suggestion! To evaluate perceptual quality, we computed the Deep Perceptual Audio Metric (DPAM)  on the audio clips rendered for user study and demo. These clips were generated by convolving predicted RIRs with dry sound and comparing them to ground-truth renderings.
>
> Versa-ELE reduces DPAM error by 20%, and Versa-SSL further reduces error by 36% compared with Versa-ELE on the AVR method. Versa-ELE also boosts the performance of other methods by 18%.  These show our method can improve the perceptual similarity to ground truth. These results are consistent with the user study findings and reinforce the perceptual benefit of our method in the resounding task. We will include this new metric and its analysis in the revised version.
>
> |        Method       |  DPAM score  |
> |:-------------------:|:------------:|
> |   NAF + Versa-ELE   | 1.53 -> 1.26 |
> |  INRAS + Versa-ELE  | 1.42 -> 1.19 |
> | AV-NeRF + Versa-ELE | 1.47 -> 1.15 |
> |   AVR + Versa-ELE   | 1.44 -> 1.17 |
> |   AVR + Versa-SSL   | 1.44 -> 0.86 |
>
> ---
> > **Review Q3: Evaluation of Versa on BEE.**
>
> We thank the reviewer for the follow-up question regarding the comparison to BEE and the potential for combining Versa-ELE with that framework. We address this question as follows:
>
> 1. **Different Settings and Objectives**: Echoing our response to W1-part1, BEE and our work target distinct problem settings. BEE is designed for cross-scene generalization: it is trained on a large-scale, multi-scene dataset with dense emitter-listener pair sampling, and it leverages audio-visual correspondence to model the acoustic field. After large-scale training, it can be adapted to novel scenes using sparse listener observations. Versa is specifically designed to improve generalization of neural acoustic fields within a single scene, leveraging reciprocity as a physics prior for improved modeling of wave propagation. We also contain only a few emitter locations training dataset, instead of densely sampled emitter-listener pairs in lots of scenes as in BEE.
>
> 2. **Evaluation on other related models**: While we do not compare directly to BEE, we evaluate Versa on other recent open-source neural acoustic field methods that also use audio-visual correspondence, including AVCloud and NeRAF/AV-NeRF. These methods operate within a single scene and use visual priors for field reconstruction.  Our results demonstrate that Versa is compatible with such architectures and consistently improves performance.

---

> > ### Comment · Reviewer_eBxj · 2025-08-06
> > **Thank you to authors for the rebuttal**
> >
> > I would like to thank the authors for the rebuttal and clarifying the points that I have raised. The additional results and descriptions are very helpful but I still have several questions and comments.
> >
> > The original submission lacked relevant works so additional results and description of the work in light of SOTA methods in acoustic filed modeling improved relevance and validation. The results provided in a new table show consistent contribution of ELE to SOTA baselines though some additional interpretation appears to be needed. I am wondering if the authors can interpret/analyze what would be the contribution of ELE to a particular approach, i.e it is surprising that AVR+ ELE reduces STFT to lower values than AVCloud+Versa-ELE, although AVCloud performs better than AVR on that metric.
> >
> > Thank you for clarifying the difference between Versa-ELE and ELE and BEE. While I agree that the training is different, my question was asking how your methods apply in the generalization to other scenes and whether there is benefit in training on variety of scenes to model acoustic field of a new given scene.
> >
> > Thank you for clarifying the exact expression of contrastive loss. I am wondering if the more common way of defining contrastive loss could be done in your setup and how it will impact the results.
> >
> > Regarding number of scenes, I would like to point out that it would add more confidence if the method would be validated on a variety of scenes and results would be reported as average and standard deviation. Testing on 9 real + 6 simulated scenes (as far as I understand that the authors used ) is just does not appear as extensive evaluation.

---

> > > ### Author Response · Authors · 2025-08-07
> > > **Further reply part 1**
> > >
> > > (This is the part one of the further reply)
> > >
> > > We sincerely thank you for the thoughtful follow-up. Your detailed comments and questions have further helped us clarify our contributions and identify directions for refinement. We respond point by point below.
> > >
> > > **Interpreting the effectiveness of Versa-ELE across methods**
> > >
> > > We appreciate your observation on the comparative results of AVR+Versa-ELE and AVCloud+Versa-ELE. The key difference lies in how these baseline models leverage acoustic principles. AVR directly models the acoustic field through acoustic volume rendering and incorporates physical wave propagation priors in its formulation. We also observe that it is data hungry to train a reliable neural acoustic field. As such, it is more receptive to additional physically inspired constraints like reciprocity. Versa-ELE aligns well with AVR’s inductive bias and further improves its convergence and accuracy. On the contrary,  AVCloud primarily relies on learned audio-visual correspondence for acoustic field modeling and especially on the environmental anchor points for acoustic rendering. Despite this, Versa-ELE still yields significant improvements on AVCloud, showcasing the generalizability of our reciprocity-inspired learning strategy. This also validates that even visually conditioned neural acoustic field methods benefit from our inductive bias, although the magnitude of improvement may vary based on the model's underlying assumptions.
> > >
> > > **Generalization to Novel Scenes and Training on Scene Variety**
> > >
> > > We appreciate the reviewer’s consideration on generalization. While our current paper focuses on per-scene optimization and does not directly address cross-scene generalization, we agree this is an important and promising direction. Currently, most of the novel scene acoustic reconstruction work primarily relies on the visual-acoustic correspondence [R1, R2, R3]. In our manuscript, the experimental results show that methods relying on visual priors (e.g., AV-NeRF, AVCloud, NeRAF) benefit from Versa. This suggests that our reciprocity-inspired training strategy can complement visual conditioning and potentially assist in large-scale generalization. We envision two scenarios where Versa would remain useful in multi-scene generalization setups:
> > > - Training phase: many novel-scene generalizations methods involve training RIRs in lots of scenes. Versa can be used in this phase to augment or regularize the learning using symmetry prior, thus improving generalization with potential limited supervision. This is valuable since collecting real-world RIRs at many different scenes is expensive.
> > > - Inference phase: During the inference phase, recent work also needs few extra RIRs as reference in the scene to generalize to this new environment. One can also use Versa to boost the reference samples to help the model to generalize to a novel scene.
> > >
> > > We therefore believe that reciprocity-inspired learning not only improves single-scene reconstructions, but also offers meaningful advantages in multi-scene training regimes when incorporated as part of physics-informed learning. We will elaborate on this point in the discussion section and include relevant papers [R1, R2, R3] in the final manuscript.
> > >
> > > [R1] Chen et al.,  "Be everywhere-hear everything (bee): Audio scene reconstruction by sparse audio-visual samples." CVPR’23.
> > >
> > > [R2] Liu et al., “Hearing Anywhere in Any Environment” CVPR’25
> > >
> > > [R3] Chen et al., "SoundVista: Novel-View Ambient Sound Synthesis via Visual-Acoustic Binding." CVPR’25.

---

> > > > ### Author Response · Authors · 2025-08-07
> > > > **Further reply part 2**
> > > >
> > > > **On the Use of Common Contrastive Loss for Versa-SSL**
> > > >
> > > > Thank you for raising this important question. While contrastive learning frameworks typically rely on both positive and negative pairs, we found this formulation less effective in our setting. We empirically tested contrastive losses on our task. We define positive samples as emitter/listener-swapped RIR pairs and treat the remaining pairs as negatives. We found this would introduce little performance improvement. This is due to the fine-grained nature of neural acoustic field modeling, which requires precise mappings between emitter-listener pose pairs and their corresponding RIRs. Unlike a simple example in pattern recognition tasks like a dog and a cat image clearly forming a negative pair, it is challenging to define meaningful negative pairs for acoustic fields. Two different emitter/listener configurations may still produce highly similar RIRs if the room geometry results in similar acoustic propagation paths. Conversely, even emitter/listener pairs in close proximity may yield drastically different RIRs due to occlusions or structural boundaries such as walls. As a result, similarity based purely on spatial distance is not a reliable criterion. To summarize, we believe designing contrastive self-supervised learning tailored to acoustic field learning could be a promising direction and we will discuss this in the revised manuscript.
> > > >
> > > > |      Variant     | STFT |  Amp |  Env |  T60 |  C50 |  EDT |
> > > > |:----------------:|:----:|:----:|:----:|:----:|:----:|:----:|
> > > > |      Vanilla     | 2.58 | 0.89 | 3.97 | 52.8 | 5.38 | 52.4 |
> > > > | Contrastive loss | 1.78 | 0.47 | 3.45 | 25.3 | 3.22 | 36.0 |
> > > > |     Versa-SSL    | 1.32 | 0.26 | 3.10 | 16.8 | 1.82 | 22.4 |
> > > >
> > > > **Scene Diversity and Reporting Aggregated Metrics**
> > > >
> > > > We appreciate the reviewer’s emphasis on broader scene validation. Due to time constraints of the rebuttal phase, we are unable to provide extra experiments on all our baseline models with  Versa within the remaining two days. We acknowledge the importance of evaluating across more scene settings and will report results with mean and standard deviation across a broader set of scenes in the camera-ready version. However, we believe our current evaluation still demonstrates the robustness and effectiveness of Versa. Our evaluation settings in the original manuscript span a variety of scenes with various reverberation duration, listener/emitter placements, number of listener/emitter samples in the training dataset, rich geometries layout and scene shapes. We hope this breadth can provide meaningful evidence of our method’s broad applicability and encourage further validation in future work.
> > > >
> > > > **Summary with Proposed Manuscript Revisions:**
> > > >
> > > >  Again, we sincerely thank you for the constructive feedback, which has substantially helped us improve the scientific clarity and scope of our paper. To address your key concerns, we will make the following revisions in the final manuscript:
> > > > - Expand the discussion in the results section to analyze the varying impact of Versa across different backbone models (e.g., AVR vs. AVCloud), highlighting the influence of model design and reliance on acoustic priors.
> > > >
> > > > - Add a new discussion paragraph to explain how Versa can be applied or extended to multi-scene generalization settings, including during both training (data augmentation with symmetry priors) and inference (enhancing limited RIR reference data).
> > > >
> > > > - Revise the Related Work section to include missing recent baselines (e.g., NeRAF, AVCloud, INRAS++), and clarify the distinction between our setting and works like BEE. Include citations to recent works on cross-scene acoustic field generalization that use visual-acoustic correspondence, and position Versa as a complementary approach in such pipelines.
> > > >
> > > > - Clarify why standard contrastive loss is not well-suited for our task, and report its performance empirically compared to Versa-SSL, with a new results table added in the Appendix.
> > > >
> > > > - Acknowledge the reviewer’s suggestion on averaging across more diverse scenes. We will include mean and standard deviation across scene configurations in the camera-ready version, and state this intent in the paper.
> > > >
> > > > We believe these revisions will significantly strengthen the manuscript and provide clearer context and utility for our proposed framework.

---

> ### Author Response · Authors · 2025-08-08
>
> Dear reviewer,
>
> We would like to thank you for the time and constructive feedback during the review process. We have addressed each of your further comments in detail and proposed corresponding revisions to the manuscript. If you have any further questions, clarifications, or suggestions, we would be happy to address them before the discussion period concludes. Thank you again for your engagement. Your suggestions are highly valuable and would greatly help us improve our work.

---

> > ### Comment · Reviewer_eBxj · 2025-08-08
> >
> > Thank you to authors for providing additional (enlightening!) results re. my comments on contrastive loss, discussion of generalization, and overall summary of revisions. As a result of the rebuttal and discussion I have more understanding and confidence in reciprocity-inspired learning and I'm more positive leaning toward this work. I'd encourage the authors to incorporate the revisions presented in the rebuttal in the revised version of the paper.

---

> ### Author Response · Authors · 2025-08-08
>
> Dear reviewer,
>
> Thank you for your positive feedback and for recognizing our method's reciprocity-inspired learning strategy. We will update the manuscript according to all your suggestions. We hope our revisions and the demonstrated results further justify your support for our paper's acceptance.

---

### Official Review · Reviewer_9Edj · 2025-07-04

**Clarity:** 3
**Significance:** 2
**Originality:** 3
**Rating:** 5
**Confidence:** 4

**Summary:**

This paper introduces the task called "resounding", which consists in regenerating a sound field for a new emitter position, based on recordings of a sparse set of emitter locations. Although this is a particular case of the well-studied task of room-impulse-response/Acoustic-transfer-function interpolation, it has been relatively less studied than interpolating between different *receiver positions*. One difficulty is that available datasets generally have less emitter positions available. To counter this, the authors propose a simple trick: use the well-known reciprocity of the wave equation to perform data augmentation by swapping the roles of the emitter and receiver. While this is straightforward in the case of omnidirectionnel emitters and receivers, the authors tackle the trickier case of directive devices by including a joint emitter-source response as an input of the neural acoustic field, and train it to learn propagation path responses that are invariant to swapping both angles of arrival/departure *and* emitter/receiver positions, via self-supervised learning. The authors show that this trick enables a boost in performance for a number of existing state-of-the-art neural acoustic field architecture.

**Questions:**

- My main interrogation is as written above: is this really beneficial compared to simple making the neural acoustic field invariant to permutation of the relevant input?

- Eq. (8): Is G really fed twice to the neural acoustic field F? Why? Please clarify to avoid confusion.

- L200: "As a result, Versa-SSL does not generate new valid training samples with ground-truth impulse response as Versa-ELE." something seems wrong ith this sentence, I don't understand.

- I am aware of the following work that also utilizes the reciprocity principle in the context of interpolating acoustic transfer functions (the submitted paper cite other works from these authors related to PINNs, but I am not sure those use reciprocity):  Ribeiro, J. G., Koyama, S., & Saruwatari, H. (2023). Kernel interpolation of acoustic transfer functions with adaptive kernel for directed and residual reverberations. In IEEE International Conference on Acoustics, Speech and Signal Processing (ICASSP) (pp. 1-5). IEEE.

**Typos/suggestions**
- L96: extra space before coma
- L118: I would suggest to remove "for simplicity" here, which triggers confusion, and write instead "We consider first....", to make clear that the multipath case will be treated just after.

**Ethical Concerns:**

["NO or VERY MINOR ethics concerns only"]

**Final Justification:**

I thank the authors for their response which helped clearing my doubts on originality and significance. I am consequently improving my rating from 4 to 5.

**Limitations:**

yes

**Quality:**

3

**Strengths And Weaknesses:**

**Strengths**
- The paper is well presented and motivated
- Although the idea is simple, it is well formalized and executed, in particular for the case of directive devices
- The approach offers performance boosts to a number of existing methods
- The studied task is relevant

**Weaknesses**
- At least for the case of omnidirectional devices, I have a hard time believing that the data-augmentation trick of swapping source and receiver based on reciprocity has not been proposed before, since this is a very well known property. This makes me question the originality, although to be honest I can't pin point a reference for this. And the fact that the more interesting case of directive devices is considered somewhat counter-balances this.
- It is relatively easy to make any neural network architecture independent of permutation of two input, for example by passing each input through the same layers, and then sum the resulting embeddings. How would such an approach compare to the proposed data augmentation strategy applied to a generic architecture? This makes me question the significance of the approach

---

> ### Author Rebuttal · Authors · 2025-07-29
>
> We thank Reviewer 9Edj for the careful review and constructive suggestions. The constructive feedback has significantly contributed to the improvement of our paper. We are glad that the reviewer found our work “well presented and motivated” and “well formalized and executed”, giving us a positive rating. We address your questions and concerns as follows.
>
> ---
> > **Review W1: Concern about the paper's originality on the reciprocity theory.**
>
> We appreciate the reviewer’s point. While reciprocity is a fundamental property in the wave propagation theory. To our knowledge, no prior work has explicitly applied emitter/listener swapping as a data augmentation strategy for training neural acoustic field models. Versa-ELE is effective and novel in its systematic application and broad compatibility across models to solve the resounding task. Moreover, our key contribution lies in extending beyond the same gain pattern case with Versa-SSL, which handles directional asymmetries through a physically grounded self-supervision learning. We will clarify this distinction in the revised manuscript.
>
> ---
> > **Review W2, Q1: Question on the permutation invariant structure.**
>
> Our method is grounded in the physical property of acoustic reciprocity. The impulse responses remain unchanged when emitter and listener positions are swapped, but only under certain conditions, such as omnidirectional gain patterns. This condition is important when evaluating whether architectural permutation invariance is an appropriate solution.
>
> 1. **Discussion on permutation invariance architecture:** Based on the reciprocity principle, it is possible to design permutation-invariant neural networks (e.g., with shared encoders for emitter/listener poses), such architectures enforce symmetry unconditionally, regardless of whether impulse response remains the same with direct swapping. However, emitters and listeners can have different directional gain patterns, which breaks the impulse response invariance (discussed in the sec3.3). Enforcing symmetry in such cases can encourage the model to make incorrect assumptions. This is also the motivation for the development of our proposed Versa-SSL in the case of different emitter/listener gain patterns. Versa-SSL shows improved performance in these cases that simple permutation invariant networks cannot achieve.
>
> 2. **The advantage of Versa**: Versa-ELE explicitly leverages reciprocity when it holds by generating physically valid virtual samples through pose exchange. This strategy is effective, model-agnostic, and compatible with any architecture, which does not require changing the model architecture. Furthermore, Versa-SSL generalizes this idea to asymmetric gain patterns by introducing self-supervised consistency constraints, without assuming symmetry.
>
> 3. **Additional Experiment results**: We provide experimental results (on AcoustiX-Diff with different gain patterns) on a permutation-invariant version of NAF. As shown in the table, permutation-invariant structure can improve the performance on the original NAF model under the sparse emitters settings. As this design treats the emitter/listener exactly the same, it can alleviate the problem of sparse emitters. However, the permutation-invariant design strictly forces the model to output symmetric results and does not consider the unique features of the emitter or listener, i.e. different gain patterns. In contrast, vanilla neural acoustic models that encode emitter/listener features separately are not strictly constrained to this inappropriate symmetry and can benefit from the Versa-ELE for better performance. Finally, AVR with Versa-SSL achieves the best overall performance as it correctly handles the different gain patterns.
>
>
> |                   | STFT |  Amp | Env |  T60 |  C50  |  EDT |
> |-------------------|:----:|:----:|:---:|:----:|:-----:|:----:|
> |        NAF        | 6.51 | 0.68 | 4.1 | 45.6 |  5.49 | 42.3 |
> | NAF + permutation | 4.47 | 0.62 | 4.0 | 25.9 |  3.90 | 37.9 |
> |  NAF + Versa-ELE  | 4.24 | 0.58 | 3.9 | 15.1 |  3.67 | 30.9 |
> |  AVR + Versa-SSL  | 1.32 | 0.26 | 3.1 | 16.8 | 1.852 | 22.4 |
>
>
> ---
> > **Review Q2: Confusion on equation 8: $F(P_l, P_e, w_l, w_e, G, G) = F(P_e, P_l, w_e, w_l, G, G)$**
>
>
> G is used as both the emitter gain pattern and the listener gain pattern, thus appearing twice in the equation. In the implementation, the emitter gain pattern $G_e$ is implicitly encoded in the neural acoustic field, and it is not available to be changed. Thus, we can only align the listener pattern with $G_e$, forcing the acoustic volume rendering step of AVR to use the same listener gain pattern as the implicit emitter pattern. As a result, we only alter the microphone pattern (we replace the microphone pattern with $G_e$ during the acoustic volume rendering) in the self-supervision stage.
>
> ---
> > **Reivew Q3: Clarification on the L200: As a result, Versa-SSL does not generate new valid training samples with ground-truth impulse response as Versa-ELE.**
>
> Sorry for the confusion in this sentence. For Versa-ELE, it is implemented as a data augmentation method to generate valid impulse response training samples. For Versa-SSL, it forces the neural acoustic field to output consistently after exchanging the emitter/listener poses in a self-supervised way.
>
> ---
> > **Review Q4: Discussion of the relevant paper.**
>
> We thank the reviewer for pointing out the work by Ribeiro et al. (ICASSP 2023). While that paper discusses reciprocity in the context of kernel interpolation and physics-informed modeling, our approach differs in both scope and methodology. No prior work has leveraged reciprocity as a systematic training strategy for the neural acoustic field. Our contribution lies in using reciprocity to improve model generalization in sparse emitter settings, and in extending this idea to asymmetric scenarios through Versa-SSL. We will update the related work section to better clarify this distinction.
>
> ---
> > **Typos/Suggestions.**
>
> Thank you for your detailed suggestions on the writing and text. We will incorporate them for the revised version after rebuttal.

---

> > ### Comment · Reviewer_9Edj · 2025-08-05
> >
> > I thank the authors for their response which helps clearing my doubts on originality and significance. I am willing to improve my rating from 4 to 5.

---

> > > ### Author Response · Authors · 2025-08-05
> > >
> > > We are glad that our response can clear your concern about our originality and contributions. Thank you for your active discussion and we will revise the manuscript according to your suggestions!

---

### Note · Authors · 2025-08-12

Dear AC and Reviewers:

Thank you for your active engagement during the rebuttal and discussion phases, which helped improve the clarity, rigor, and impact of our work. Below we summarize each reviewer’s key feedback and final opinion.

- **Reviewer 9Edj** found the work “well presented and motivated” with clear performance gains across methods. During the rebuttal, we presented additional results which showed that simple permutation-invariant structures were less effective than our approach, addressing the reviewer’s concerns on originality. **The reviewer was willing to raise the score from 4 to 5.**

- **Reviewer eBxj** noted that our work addresses an “important problem” and can be combined with existing methods to improve multiple metrics, but requested broader related-work discussion and more test scenes. We added generalization results to other recent works and more analysis of our formulations, which **the reviewer found it “enlightening” and had “more confidence in our reciprocity-inspired learning” and “more positive leaning toward this work”.**

- **Reviewer tDBR** valued the “combination of physical modeling sophistication with machine learning sophistication” but was concerned about potential impact. We expanded discussion on applications within and beyond the audio community, highlighting applicability to other wave-based domains. **The reviewer appreciated these clarifications, which made the paper “stronger as a result”.**

- **Reviewer sQL3** praised our focus on a “specific challenge” and “simple technique” to be effective across diverse datasets, but questioned the validity of reciprocity across datasets. We provided additional real/simulated validations, clarified assumptions, and justified reciprocity as a learning inductive bias for acoustic fields. **The reviewer “appreciate our detailed responses” and raised the score from 3 to 4.**

**Summary**: Reviewers expressed increased confidence and appreciation for our contributions after discussion. Two reviewers stated raising scores explicitly, others provided suggestions for manuscript strengthening and expressed more positivity towards the work, and all acknowledged the value of our reciprocity-based learning framework and its demonstrated improvements. The discussion phase confirmed that our work is well-motivated, broadly applicable, and contributes both practical performance gains and conceptual advances in physics-guided neural acoustic field modeling.

---

### Decision · Program_Chairs · 2025-09-17

**Decision:**

Accept (poster)

**Comment:**

The submission initially received mixed ratings, with two reviewers leaning toward acceptance and two toward rejection. Reviewers found the problem of estimating the acoustic field at novel emitter positions (resounding) interesting, and they noted that the proposed solution is simple, grounded in established principles, and demonstrates improved performance across multiple datasets compared to baselines.

However, concerns were raised regarding the clarity of some technical aspects, missing related work, and the relevance of the proposed task. The authors rebuttal effectively addressed these issues, and most reviewers stated that their concerns were largely resolved. Following the rebuttal, all reviewers recommended acceptance.

The AC concurs with the reviewers consensus and recommends acceptance of the submission. The authors are encouraged to incorporate their rebuttal and the reviewers’ feedback into the final version.